# Curating the Future: A Scalable Recipe for Training Open-Ended Forecasters

**Nikhil Chandak** [* 1 2]  **Shashwat Goel** [* 1 3]  **Ameya Prabhu** [† 2 4]  **Moritz Hardt** [† 1 2]  **Jonas Geiping** [† 1 3 2]

🌐 Blog    Code    🗄 Dataset & Models

## Abstract

High-stakes decision making involves reasoning under uncertainty about the future. In this work, we train language models to make predictions on open-ended forecasting questions. To scale up training data, we synthesize novel forecasting questions from global events reported in daily news. While directly training on this data leads to performance drops, carefully curating questions creates a valuable training resource. We use the resulting dataset, `OpenForesight`, to post-train Qwen3 thinking models. To prevent leakage of future information during training and evaluation, we use an offline news corpus, both for data generation and retrieval in our forecasting system. Guided by a small validation set, we show the benefits of retrieval, and an improved reward function for reinforcement learning (RL). Our specialized model, `OpenForecaster`-8B, matches much larger proprietary models on held-out testing, with our training improving both accuracy and calibration of predictions. We find calibration improvements from forecasting training generalize across popular benchmarks.

## 1. Introduction

Every day, people navigate decisions under uncertainty, due to incomplete evidence or competing hypotheses. The highest-stakes choices are inherently forward-looking: governments set policy while anticipating macroeconomic and geopolitical shifts; investors allocate capital amid market

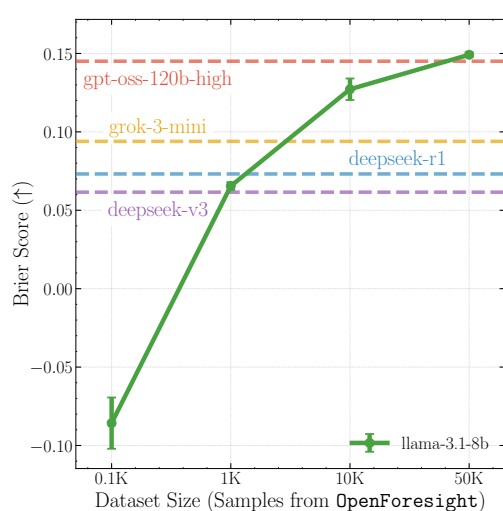

*Figure 1.* **Benefits of scaling forecasting training data.** We find the Brier score improves consistently as we increase the size of the training dataset at equal compute.

and regulatory uncertainty; and scientists pursue research directions in search of the next breakthrough. Decades of work (Tetlock et al., 2014) on human forecasting shows that while prediction is hard and skill varies widely, it is possible to train humans to become better forecasters. In fact, some "superforecasters" consistently outperform peers. However, forecasting such open-ended outcomes is still difficult: while there is a ceiling to predictability in social systems (Franklin, 1999), we do not yet know where it lies.

If trained at scale for forecasting world events, Large Language Models (LLMs) may enjoy structural advantages over humans: they can ingest and synthesize vast, heterogeneous corpora across thousands of topics; and update predictions rapidly as new information arrives. Just like language models have shown superhuman reasoning on exam-style math and coding problems (OpenAI, 2025), in the future, language model forecasters may be able to come up with possibilities that humans miss. So in this work, we study:

> *How can we train language models to*
> *better forecast open-ended questions?*

---

[*]Equal contribution [†]Equal co-supervision [1]Max Planck Institute for Intelligent Systems, Tübingen, Germany [2]Tübingen AI Center, Germany [3]ELLIS Institute Tübingen, Germany [4]University of Tübingen, Germany. Correspondence to: Nikhil Chandak <nikhil.chandak@tuebingen.mpg.de>.

*Proceedings of the 43[rd] International Conference on Machine Learning*, Seoul, South Korea. PMLR 306, 2026. Copyright 2026 by the author(s).

**Scaling training data for forecasting.** As forecasting world events is hard for humans, detailed and correct reasoning traces for forecasting are difficult to obtain. Fortunately, recent success in Reinforcement Learning (RL) for language models enables training with just the eventual outcome of the question (Guo et al., 2025). Further, the static knowledge cutoff of LLMs creates a unique opportunity: events that resolve after the cutoff are in the future for the model. Even then, sourcing questions at scale for training forecasting abilities has a few key challenges. First, waiting for events to resolve is too slow as a feedback loop for training. Second, prediction markets–the primary source for existing forecasting questions–rely on humans to come up with forecasting questions which is not scalable. Moreover, they mostly consist of binary yes or no questions, where a 50% chance of success even with incorrect reasoning makes for noisy rewards.

We instead turn to global news, which covers a large number of salient events every day, to synthesize open-ended forecasting questions like "Who will be confirmed as the new prime minister of Ukraine on 17 July 2025?". Our recipe for creating training data is entirely automated and scalable, with one language model extracting events from news articles to generate questions, and a different model filtering and rewriting questions to avoid leaking future information. For this work, we use this recipe with 250,000 articles up till April 2025, to create `OpenForesight`, a dataset of $\sim$ 52,000 open-ended forecasting questions for training. Figure 1 shows the benefits of scaling training data for our RL recipe, when applied to Llama 3.1 8B, which starts without RL training for reasoning. To grade responses for open-ended questions, we use model-based *answer matching* (Chandak et al., 2025) consistent with frontier benchmarks like Humanity's Last Exam (Phan et al., 2025).

**Our data curation leads to a qualitative difference.** Creating realistic forecasting questions from news can deceptively hard as news reports events which have effectively been decided. We find naive question generation often leaks the outcome through wording, phrasing of background context, or resolution criteria. Such leakage creates easy shortcuts and training with RL on such questions *worsens* forecasting performance. Our pipeline therefore performs extensive filtering and targeted rewriting to ensure questions are forward-looking and free of answer-revealing clues. This helps turn noisy news-derived questions into a reliable training signal. This curation makes the whole difference: without it, training hurts forecasting, whereas with it, performance increases sharply (Figure 3).

**Ensuring we truly improve forecasting.** We take extensive measures to avoid the leakage of future information during training and evaluation. First, we do not use online search engines for sourcing news, as they have unreliable date cutoffs due to dynamic updates to documents and search ranking (Paleka et al., 2025a). Instead, we use the CommonCrawl News corpus, which provides static, monthly snapshots of global news. Second, we only train on events until April 2025, which is when the Qwen3 model weights we train were released. Finally, we do not observe performance on the test set until the very end. Our test set is composed of diverse news sources, different from the ones used in training and validation, to ensure we are not just learning distributional biases of the training data.

**Design choices when training LLM forecasters.** We start from the Qwen3 (Yang et al., 2025) 8B model with thinking enabled. We perform all ablations on a small validation set. We use the Qwen3-8B Embedding for retrieval, providing forecasters the most relevant chunks from our offline news corpus. The retrieved information leads to large improvements, despite a cautious approach of only retrieving articles until *one month* before the question resolution date to avoid leakage. Then, we train language models using RL with GRPO. For the reward function, we propose combining accuracy, and an adaptation of the brier score for open-ended responses (Damani et al., 2025). Ablations show rewarding accuracy alone hurts calibration, while optimizing only the brier score hurts exploration on hard questions.

**Final results.** In Section 6, we report results on our held-out test set of open-ended forecasting questions from May to August 2025, and FutureX (Zeng et al., 2025), an external forecasting benchmark. RL training on `OpenForesight` makes the predictions of our specialized 8B model competitive with much larger proprietary models in both accuracy and calibration. We also observe large improvements on consistency evaluations for long-term predictions (Paleka et al., 2025b). Finally, we find calibration from our forecasting training generalizes to multiple out of distribution benchmarks.

By providing rigorous probabilistic predictions, open-ended forecasting systems could transform policy making, corporate planning, and financial risk management (Tetlock, 2017). To promote forecasting research, we open-source our dataset, code, and models.

## 2. Related Work

**Forecasting World Events.** Much prior work in Machine Learning and Statistics has focused on forecasting numeric or time-series data (Box & Jenkins, 1976) in diverse domains like weather (Richardson, 1922), econometrics (Tinbergen, 1939) or finance (Cowles, 1933). Instead, our work focuses on the prediction of discrete world events, with both questions and answers described in natural language, also called *judgemental forecasting* (Tetlock

& Gardner, 2016). In the rest of our paper, we refer to this as *forecasting* for brevity. In prior work on evaluating language models for forecasting (Zou et al., 2022; Karger et al., 2024), questions are primarily sourced from prediction markets like Metaculus, Manifold, and Polymarket. Prediction markets provide a platform for online participants to register predictions on questions like "Will Donald Trump win the US Presidential Election in 2024?", which mostly have binary, yes or no, outcomes and have rapidly grown in popularity over the last few years.

**Evaluating LLMs for Forecasting.** New information (before the event resolves) benefits forecasting. Thus, LLM forecasting work (Zou et al., 2022; Halawi et al., 2024) provides relevant retrieved articles to models (Lewis et al., 2020) often obtained via web-search APIs. Paleka et al. (2025a) discuss pitfalls of LLM forecasting evaluations, including leakage of outcomes from online search in backtests, and distributional biases of prediction market questions. To avoid these issues, we use static, monthly snapshots of global news for retrieval and creating questions. Jin et al. (2021) ask humans to create forecasting questions, while Dai et al. (2024) try to automate this process with LLMs. However, their questions pre-define a few outcomes to choose from. Guan et al. (2024); Wang et al. (2025) evaluate models on open-ended forecasts, but we go a step further by showing how to train models for this task.

**Reinforcement Learning for LLMs.** Shao et al. (2024) proposed *Group Relative Policy Optimization* (GRPO), an RL algorithm that only uses outcome rewards. This approach has been highly successful in training LLMs to *reason* about well-specified coding (Jain et al., 2024) and exam-style questions across domains (Phan et al., 2025). Instead, forecasting requires LLMs to reason about uncertainty. Halawi et al. (2024) proposed training language models for forecasting by Supervised Finetuning (SFT) on chain of thought traces that lead to brier scores better than the prediction market aggregate. In the same setting of binary forecasting questions, Turtel et al. (2025a) optimize brier scores using GRPO, while Damani et al. (2025) extend it to short answer questions in other domains. We depart from these works in showing how to synthesise large-scale forecasting training data from daily news, to train models at open-ended reasoning about the future.

## 3. Open-Ended Forecasting

**Motivation.** The forecasting task we study is *open-ended* in two key ways: 1) It allows expressing arbitrary natural language forecasting questions 2) It may not have a structured outcome set, unlike numeric or categorical predictions. This differentiates it from both time-series forecasting, and prediction markets. For example, prediction markets are dominated by binary (yes/no) or multiple choice questions. While this design is easy to score the most important questions to forecast—such as scientific breakthroughs, geopolitical shocks, or technological disruptions—often emerge as *unknown unknowns*: possibilities not anticipated, and hard to enumerate. Thus, in this work, we focus on training models to make open-ended predictions like "Which tech company will the US Government buy a >5% stake in by September 2025?". Such questions require exploration and imagination, rewarding novel hypotheses that turn out to be correct, rather than just distributing probabilities over a known set of outcomes.

**Background.** LLM weights are frozen after training, especially for open-weight models. Any event that happens after the last date in the training corpus is in the future for the LLM. This provides a time window to collect questions for training models to reason about future events. Similarly, their evaluation involves testing on questions resolving after the cutoff date of the training data, called *backtesting* (Tashman, 2000). While prior work has relied on prediction market questions as training data, this has three key problems:

1. The questions are created by humans, which makes them low in number (Paleka et al., 2025a). This becomes a bottleneck for scaling training data, which has been an essential component in the success of LLMs (Kaplan et al., 2020; Lu, 2025).

2. Most questions have binary outcomes, which creates a 50% baseline success rate. This leads to noisy rewards in outcome-based RL, which means even incorrect reasoning has a high chance of being reinforced.

3. Each platform has a skewed distribution of events. All overrepresent US political news, along with their specific focus such as crypto-currency in Polymarket, technology in Metaculus, personal life of users in Manifold, and sports events on Kalshi (Paleka et al., 2025a).

These limitations motivate us to explore alternate ways to create forecasting questions.

**Setup.** Let $\mathcal{X}$ be the set of open-ended forecasting questions; and $\mathcal{Y}$ the set of short textual answers. We provide a language model $\pi_\theta$ a question $x \in \mathcal{X}$, for which we already know the ground-truth outcome $y^\star$ as it has resolved in the real-world. We ask the model to report its best guess prediction $y$, and the probability $q$ of it being the true outcome.

**Measuring Accuracy.** We measure accuracy by checking if the model's attempted answer $y$ matches with the ground truth outcome $y^\star$, using another language model to test for semantic equivalence (for example "Geoffrey Hinton" = "Geoffrey Everest Hinton") consistent with recent frontier

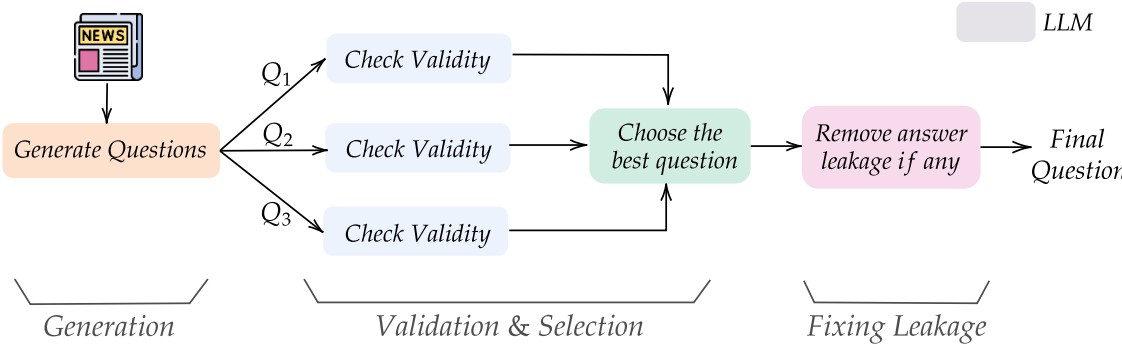

*Figure 2.* **Our question generation methodology.** We use DeepSeek-v3 to generate multiple forecasting questions per news article. Then, we use Llama-4-Maverick to check if questions follow all guidelines, choose the best question, and remove any hints revealing the answer.

benchmarks (Wei et al., 2024; Phan et al., 2025). For evaluations, we use Llama-4-Scout (Meta AI, 2025), as in a recent study (Chandak et al., 2025), it at matching answers, it has inter-human levels of alignment with human judgments. During training, we use Qwen3-4B in non-thinking mode, as it achieves high alignment levels for its size in the same study. We find the two models agree on $\sim 97\%$ grading responses, and manual validation ensures they are accurate in $\geq 95\%$ cases.

**Measuring Calibration.** We adapt the multi-class Brier scoring rule (Mucsányi et al., 2023) for free-form responses as follows (details in Section A):

$$S'(q, y, y^*) = \begin{cases} 1 - (q-1)^2, & \text{if } y \equiv y^* \\ -q^2, & \text{if } y \neq y^* \end{cases}$$

**Interpretation.** Predicting an event with a probability $q = 0$ returns a baseline score of $0$ regardless of the prediction $y$. Correct predictions receive positive scores while incorrect predictions negative. For brevity, we call $S'(q, y, y^*)$ *Brier score* throughout this paper. Our Brier score is equivalent to the reward metric used by Damani et al. (2025). They show this is a proper scoring rule, incentivizing both high accuracy and truthful reporting of probability on the answer that seems most likely. For completeness, we discuss this further in Section A.

**Training Algorithm: GRPO (Shao et al., 2024).** We train LLMs using outcome-based RL on our dataset. For each prompt $x$, we draw $K$ completions $\{(y_i, p_i)\}_{i=1}^K \sim \pi_\theta(\cdot \mid x)$ and compute rewards $r_i = R(y_i, p_i; y^*)$. However, following prior work (Damani et al., 2025; Turtel et al., 2025b), we do not divide by the group standard deviation when computing advantages. This stabilizes updates in settings like ours where reward variance can be small.

**Initial Policy: Qwen3 Thinking (Yang et al., 2025).** We start with the 8B thinking model whose knowledge–cutoff date is not officially reported. When queried directly, the

models return inconsistent cutoff dates (most often *October 2023* or *June 2024*). Usually, they treat questions about 2024 as being in the future. Since the model weights were released and frozen in April 2025, we train up to this date, and use the period between May to August 2025 for testing. In the Appendix, we show large improvements from our training on even the Llama and Gemma models in Section D.3.

## 4. Generating Open-Ended Forecasting Questions from News

We now discuss our methodology to convert daily news articles into forecasting questions using language models. Any fixed forecasting dataset loses value as newer base models with training cutoffs after the dataset was created are adopted. Thus, we first describe the general methodology which can be used in the future, and then describe the specific instantiations we used to create our training data `OpenForesight` which has questions until April 2025. We conclude by demonstrating forecasting improvements due to our data filtering steps.

### 4.1. Methodology for Generating Forecasting Questions

We generate short-answer, open-ended forecasting questions from individual news articles as illustrated in Figure 2. We describe each step in detail below:

**Sourcing Event Information.** News outlets establish global infrastructure for reporting salient events as they occur. Unfortunately, Paleka et al. (2025a) show that sourcing news via online search engines is unreliable. While search engines provide date cutoffs, future information can leak through updates to articles after the publish date, and even search engine ranking. This compromises the reliablity of backtests, and leaks future information in training, which can hurt Deep Learning models that easily overfit to spurious correlations. Fortunately, the CommonCrawl News (CCNews) Corpus (Nagel, 2016) provides static monthly snapshots of

global news with accurate dates. This makes it free and easy to obtain news articles for creating forecasting questions.

**Generating samples from documents.** Based on each news article, we ask a *sample creator* model to generate up to three diverse forecasting samples. Each sample consists of: (i) a concise question about an event with an explicit deadline (e.g., "by *Month, Year*"); (ii) brief background that provides context, or defines uncommon terms; (iii) resolution criteria that fixes a source of truth and the expected answer format; (iv) The unique answer, drawn verbatim from the article, usually short (1–3 words), non-numeric (usually a name or location); and (v) Source article link for reference, obtained from article metadata. We show an example in the box below.

---

**Sample Generated Forecasting Question**

**Question.** Who will be confirmed as the new prime minister of Ukraine by 17 July 2025?

**Background.** Ukraine's parliament is scheduled to vote to appoint a new prime minister.

**Resolution Criteria.**

- **Source of Truth:** Official announcement from the Verkhovna Rada (Ukraine's parliament) confirming the appointment, via parliamentary records or government press release.

- **Resolution Date:** 17 July 2025, the date on which the parliamentary vote occurs and results are published.

**Answer Type.** String (Name)

**Ground-Truth Answer.** Yulia Svyrydenko

**Source.** The Guardian (live blog): Ukraine live updates — 17 July 2025

---

A challenging issue we face is that sometimes news articles talk about past events, or report an event late. This is why we ask the sample creator to propose a resolution date, and set the final resolution date as $\min($model_generated_date, article_publish_date$)$. We perform additional steps, including manual review, to address this issue for evaluation questions, as described later in Section 6. For training data, we do not add more complex steps to fix resolution dates due to cost constraints.

**Filtering samples.** For each question, we use another LLM, *the sample selector*, to verify: (i) The question-answer pair is fully based on information in the source article (ii) The question is forward-looking, for e.g. it is in future tense

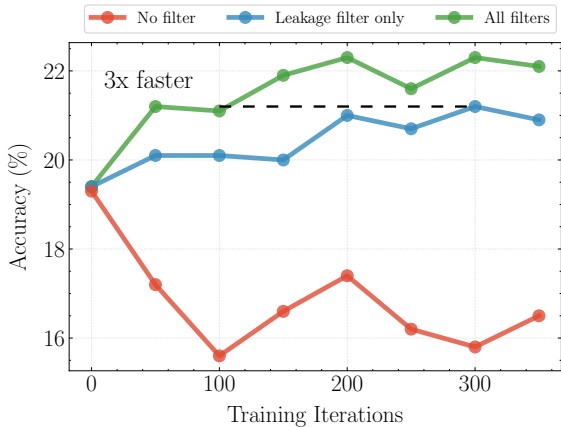

*Figure 3.* **Benefits of our filtering stage.** Without leakage removal (red), the model worsens at forecasting, possibly learning shortcuts. With only the leakage removal step (blue), we find that achieving the same performance requires $3\times$ more compute and data. Applying all filtering steps (green) leads to higher accuracy.

(iii) The answer is definite, unambiguous, and resolvable by the publication date. We mark a question as valid only if it passes these checks. If multiple questions from a single article remain, we ask the sample selector to pick the best one, favoring questions with clear, unique answers and high relevance. This is to ensure data diversity and enhance quality.

**Editing to fix leakage.** At this stage, we find that even the filtered samples sometimes leak information about the answer. This can create shortcuts during training. To fix this, we do a final editing stage where we ask the sample selector to scan the title, background, and resolution criteria to check if they reveal the answer. When it finds leakage, we ask it to rewrite only the offending spans, replacing specifics with generic placeholders. Finally, we re-scan using exact string matching any remaining mentions of the answer, and discard those samples.

Overall, this pipeline can continually ingest news articles and generate open-ended forecasting questions. We use the same methodology to create train, validation and test splits, but use *different news sources* to check if our model learns generalizable forecasting skills.

### 4.2. `OpenForesight`: An Open, Large-Scale Training Dataset for Forecasting

We now describe the specific composition of our dataset. We use DeepSeek v3 as the sample creator and Llama-4-Maverick as the sample selector, with prompts in Section G.

**Generating samples.** One practical issue we face is that many top news sources, such as The Reuters and Associated Press (AP), have disallowed scraping even for Common-Crawl, due to the rise of commercial use in language model training (Grynbaum & Mac, 2023; Longpre et al., 2025).

*Table 1.* Number of questions after each filtering stage.

| Stage | Number (% Total) |
|---|---|
| Source Articles | 248,321 |
| Question Generation | 744,963 (100%) |
| Validation | 295,274 (40%) |
| Best Question Selection | 157,260 (21%) |
| Fixing Leakage | 150,500 (20%) |
| Answer Type Filtering | 62,279 (8%) |
| Resolving after 2024-01-01 | 52,183 (7%) |
| **Final Set** | 52,183 (7%) |

Still, we are able to collect articles from popular outlets spanning diverse geographies and topics.

Particularly, for our training set, we start with $248,000$ deduplicated English-language articles between June 2023 to April 2025 from *Forbes*, *CNN*, *Hindustan Times*, *Deutsche Welle*, and *Irish Times*. The distribution is described in Figure 9. From each article, the sample creator produces three forecasting samples, yielding $745,000$ samples.

**Filtering samples.** Table 1 contains a breakdown of questions remaining after each filtering stage. 60% of question-answer candidates are marked invalid — most commonly because the article does not unambiguously resolve the question to the given answer. At this stage, zero questions remain from 40% articles, and 21% articles yield exactly one valid question, which we keep as is. For the 39% with multiple valid questions, the sample selector picks the best one. Finally, to avoid vague or numeric answers, we only keep questions with specific types (listed in Table 4).

**Editing to fix leakage.** Over $40\%$ of selected questions directly contain the answer string despite explicit prompts to avoid this. Rewriting and rejection removes ∼90% of such cases. For the remaining, we apply a string matching filter to remove direct leakage. Finally, we keep only those questions which resolve after January 1, 2024.

**Validation Set.** We use the same recipe to create a validation set of 207 questions using 500 random TheGuardian articles from July 2025. To ensure high-quality, we used `o4-mini-high`, a much more capable model than DeepSeek-v3, as the sample creator.

**Result: Filtering Improves Performance.** Figure 3 shows the effect of our filtering steps. We train Qwen3-8B using RL with identical hyperparameters on three data variants: (red) 30,000 original generated questions, without any filtering; (blue) 30,000 samples obtained after the question editing step to remove leakage; and (green) 10,000 samples sourced from Forbes and included in `OpenForesight`.

First, we observe the drastic impact of leakage in training. Training without leakage removal (red) worsens the model, perhaps due to shortcut learning. After the leakage removal steps, training improves the model (blue line). Yet, using all filtering stages (green line) leads to both higher accuracy and Brier score, in one-third the data and training steps.

In Section D.1, we also ablate the effect of training on binary-only, free-form-only, and combined binary and free-form data for `Qwen3-8B`. We find that free-form data is crucial for improving open-ended forecasting; however, training solely on freeform data does not improve performance on binary Metaculus questions. Training with both kind of questions achieves the best trade-off, so we use this for our final training runs presented in Section 6.

**Final training dataset.** Across stages, we remove $\sim 90\%$ of questions, yielding a training set of 52K samples, each drawn from a unique article. When shown the source article for each question, the Qwen3-32B model achieves 95% accuracy on these samples, confirming high validity. We release this training dataset, `OpenForesight`, to promote research on open-ended forecasting.

## 5. Prediction System

We now present intermediate results that guided the design decisions for our prediction system. We did not measure performance on the held-out test set throughout development, making decisions solely based on our validation set. We did not any find any notable difference between training in temporal order (sorted by resolution date), compared to training in a randomly shuffled order. Below we present results which show the benefits of our reward design for RL training, and retrieval system.

**Reward Design.** We compare three rewards for RL training:

1. **Only Accuracy** *(Baseline):* $R = \mathbb{1}_{y \equiv y^*}$. Vanilla success rewards are commonly used in literature on LLM RL with verifiable rewards (Guo et al., 2025).

2. **Only Brier score** *(Damani et al. (2025)):* $R = S'(q, y, y^*) = -q^2 + \mathbb{1}_{y \equiv y^*} \cdot 2q$. This incentivizes both correct predictions and calibrated confidence.

3. **Accuracy + Brier score** *(Ours):* $R = \mathbb{1}_{y \equiv y^*} + S'(q, y, y^*)$. We hypothesise optimizing the Brier score alone hurts exploration as when the model assigns a low confidence to its prediction, the correctness has a small impact on the Brier score. To fix this, we propose adding the accuracy term as well. In this case, even on hard questions, which merit low confidence, models get a large boost for correct predictions.

**Result: Accuracy + Brier reward improves performance.**

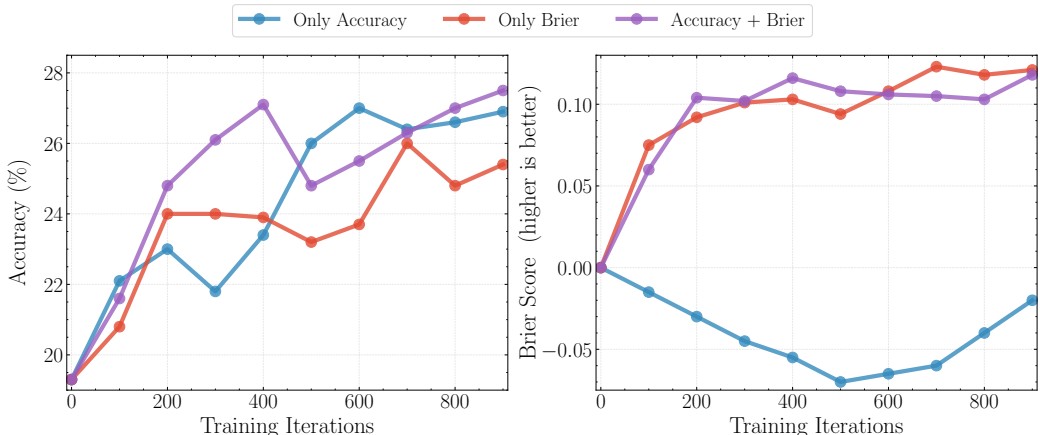

*Figure 4.* **Accuracy + Brier score reward performs the best.** Accuracy alone leads to poor calibration. While brier score incentivizes both correct predictions and calibration, on hard questions with low confidence, it provides little signal on correctness. In such cases, adding the accuracy term increases the incentive for exploration.

Figure 4 shows the validation set results of training with all three reward functions on the full `OpenForesight` dataset. We observe that optimizing accuracy alone (blue line) leads to negative brier scores, worse than a constant (0) baseline. In contrast, optimizing the Brier score alone (red line) also improves the accuracy, but to a lesser extent. Our proposed reward, accuracy + Brier (purple line), leads to the best performance on both metrics. It improves accuracy beyond the Brier alone while maintaining equal brier score on the validation set. Analyzing output distributions, we find that the brier-only trained model predicts "Unknown" with near-0 confidence in $\sim 40\%$ of samples, due to low reward for correct yet low-confidence guesses, which hurts exploration. In contrast, our proposed reward yields "Unknown" in only $\sim 4\%$ of samples, making low-confidence guesses on hard cases—improving both accuracy and Brier score.

**Retrieval.** Like prior work (Zou et al., 2022; Halawi et al., 2024), we provide the same relevant recent information across forecasting models. This provides them new evidence, or competing viewpoints to weigh, that was available before the resolution date, but potentially after the model's training cutoff. To prevent leakage issues (Paleka et al., 2025a), we use our offline CCNews corpus, and only use articles up to *one month* before the question's resolution date. Our overall pool consists of 1 million articles across 60 different sources. We de-duplicate the articles and split each into fixed-size chunks (512 tokens) and embed each chunk with the Qwen3-embedding 8B model. During evaluation, we retrieve the top-$k$ most relevant chunks and append them to the model prompt in order as context.

**Retrieval leads to large improvements in accuracy.** As shown in Section 5, providing retrieved information improves accuracy by $9-18\%$ across model families and sizes. In Appendix Figure 13, we vary the number of retrieved chunks and find that improvements plateau after five chunks.

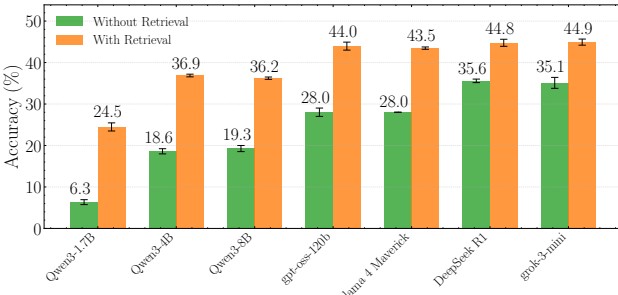

*Figure 5.* **Retrieval improves accuracy across models.** We use the specialized Qwen3 8B embedding model to retrieve the 5 most relevant chunks (512 tokens) for each question. We cautiously use articles only until a month before the resolution date.

Thus, we fix $k = 5$ chunks for all evaluations henceforth.

**Training the final forecasting system.** Based on the above design decisions guided by validation set performance, we now describe our final training methodology: We use the Qwen3-8B embedding model to retrieve the 5 most relevant chunks from news articles until a month before each question's resolution date. During training, we add a random number of such article chunks (between 0 to 5) in the prompt to make our forecaster generalizable to variable number of articles. We train the Qwen3-8B thinking model with GRPO on `OpenForesight` which has $\sim 50,000$ samples and also include 2000 binary resolved Metaculus questions from 2024. For the reward, we use Accuracy + Brier for free-form questions and only Brier for binary questions.

## 6. Final Results

We now present evaluations of our model, `OpenForecaster`-8B on held-out test sets.

**Open-ended Test Set.** Given the lack of *open-ended* forecasting benchmarks that are still "in the future" for our

models, we create our own test set with additional steps to ensure high quality. We first use our data creation recipe to generate an initial set of 1,000 questions between May to August 2025 using a stronger model, `o4-mini-high`. We draw from five diverse news sources: Al Jazeera English (global news, based out of Qatar), Time (global news, based out of USA) The Independent (UK focused), Fox News (USA focused), NDTV (India focused), with 200 questions selected from each. The choice of sources was made under the constraint of many established news sources have disallowed crawling of their articles starting 2025. We deliberately use distinct sources from the training set to ensure that our model is learning generalizable forecasting skills, and not source distribution specific biases. Beginning from this initial set of 1000 questions, we perform additional filtering steps to prepare a high-quality test set:

1. We remove any potentially unanswerable questions (noise) by keeping only those which `grok-4.1-fast` could successfully answer with search tool access (85%).

2. To address the issue of late reporting in news outlets, we again use `grok-4.1-fast` with search tool to find the **earliest resolution date** for a given question. This is important to prevent leakage from retrieving articles with the true answer. We retain only those questions with resolution date after May 2025 (64%).

3. Finally, we manually filter the remaining questions to meet our quality checks, resulting in a final test set of 302 questions. We provide more details like news source specific statistics in Section C.1.

**External Datasets.** 1) We use the FutureX benchmark (Zeng et al., 2025), filtering to non-numeric, English, resolved forecasting questions and evaluating all models with our retrieval. This leaves 86 binary or multiple choice questions, between July to August 2025. 2) For evaluating long-term predictions (without retrieval), we measure consistency metrics on binary questions up to 2028 as proposed by Paleka et al. (2025b), who show they correlate strongly with forecasting performance. 3) Finally, to measure whether our forecasting training generalizes to calibration on standard benchmarks of LLM capabilities, we evaluate without retrieval on a challenging factuality benchmark, SimpleQA (Wei et al., 2024), and popular cross-domain reasoning benchmarks, MMLU-Pro and GPQA-Diamond.

**Result: Training on our dataset leads to large improvements in forecasting.** Figure 6a shows performance of models on our held-out test set of open-ended forecasting questions. On the Brier score (Y axis), the primary metric recommended for forecasting (Tetlock & Gardner, 2016), as it measures both accuracy and calibration, `OpenForecaster`-

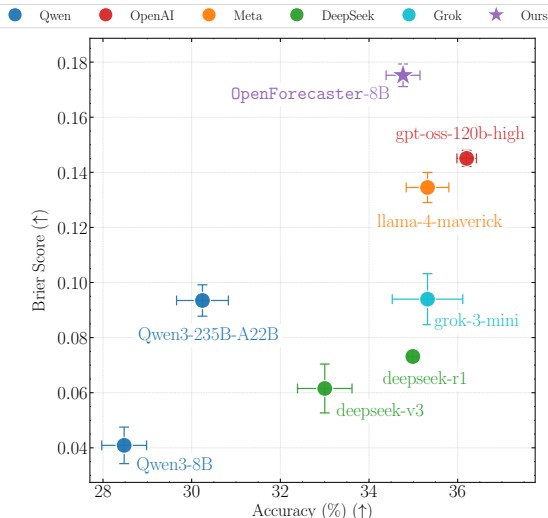

*(a)* Results on `OpenForesight` May-Aug 2025.

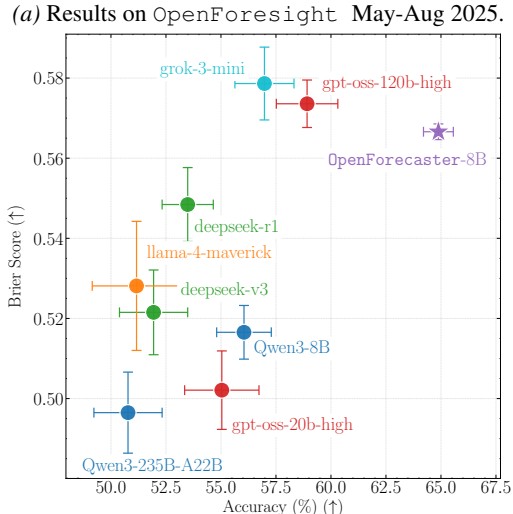

*(b)* Results on FutureX July-Aug 2025.

*Figure 6.* **Our forecasting training improves accuracy and calibration** on both, our test set, and the external FutureX benchmark. It makes `OpenForecaster`-8B competitive with much larger models that have knowledge cutoffs before May 2025.

8B significantly outperforms all other models. Our improvements are not merely from calibration; the predictions also become more accurate (X axis), with `OpenForecaster`-8B surpassing the much larger Qwen3 235B and matching models like DeepSeek-R1 and Grok-3-mini, albeit a bit behind GPT OSS 120B. We provide qualitative analysis of `OpenForecaster`-8B's reasoning improvements in Section E. Training on `OpenForesight` also improves models from other families like Llama and Gemma as we show in Appendix Section D.3. We saw a particularly large improvement (+25% absolute increase in accuracy) for Llama 3.1 8B Instruct, surpassing many larger models. We also show model accuracy by month in Figure 16.

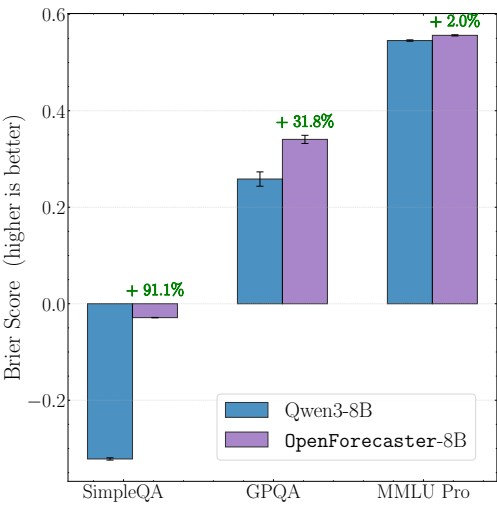

*Figure 7.* Calibration improves significantly on general benchmarks (number in green showing *relative* improvement).

| Check | Arbitrage | Frequentist |
|---|---|---|
| PARAPHRASE | -33% | -17% |
| CONSEQUENCE | -67% | -39% |
| ANDOR | -25% | -9% |
| AND | -75% | -42% |
| NEGATION | +46% | +37% |
| OR | -12% | +28% |
| BUT | -31% | -14% |
| COND | -15% | -2% |
| CONDCOND | -3% | +1% |
| EXPEVIDENCE | -18% | -19% |
| **Aggregated** | **-15%** | **-4%** |

*Table 2.* **Consistency of long-term predictions after RL training.** Percent change in violation scores (lower is better) from Qwen3-8B to `OpenForecaster`-8B.

On FutureX, our model has the strongest accuracy by a large margin, even compared to much larger proprietary counterparts. It is close to the best for Brier score as well. In Appendix Section D.2 we show our model also improves on questions from the prediction market Metaculus.

Finally, on Paleka et al. (2025b)'s dataset of long-term forecasting questions which resolve up to 2028, our training leads to more consistent long-term predictions, improving 15% on arbitrage metrics, and 4% on frequentist metrics, as shown in Table 2.

**Result: Calibration training for forecasting generalizes to other domains.** Figure 7 shows downstream improvements in calibration across SimpleQA, GPQA-Diamond and MMLU-Pro. This calibration can then be used to reduce hallucinations, for example abstaining on questions the model is not confident about, using simple rules like `if probability < 0.1, replace prediction with "I do not know."`

# 7. Conclusion

In this paper, we show how to curate data for *scalable training* of *open-ended forecasting*. The results are promising, an 8B model finetuned on our data becomes competitive with proprietary models like GPT-OSS-120B, DeepSeek-R1, and Grok-3-Mini. Calibration improvements from forecasting training generalize out of distribution. A few limitations remain. For example, we only use news to create forecasting questions, which leads to a distributional bias. The news also reports some events late, such as scientific breakthroughs, and this can make such questions easier to "predict" than others in our dataset. This should not affect relative performance comparisons between models though. We also do not consider long-form forecasts, as it is unclear how to grade these. Overall, open-ended forecasting, being a challenging and highly valuable task, offers exciting directions to pursue across research communities. A strong forecaster needs to reason about uncertainty, efficiently seek new information, and make optimal Bayesian updates to its world model. Scaling up end-to-end training of open-ended forecasting systems may lead to emergent improvements in such capabilities. By open-sourcing all our artefacts, we hope to spark more research on this important direction.

# Acknowledgments

We thank Douwe Kiela, Alexander Panfilov, Tim Rocktäschel, Yangjun Ruan, and Guanhua Zhang for valuable discussions. We thank Maksym Andriushchenko, Arvindh Arun, Alessandro Bifulco, Paras Chopra, and Daniel Paleka for helpful feedback on our draft. We thank CCNews and TheGuardian for providing free access to news articles, Thinking Machines for providing Tinker API research credits, and Contextual AI for letting us test their retrieval system.

# Impact Statement

This paper presents work whose goal is to advance the field of machine learning. There are many potential societal consequences of our work, none of which we feel must be specifically highlighted here.

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

# Appendix

## Appendix Contents

## A. Adapting Brier Score to free-form responses

Let $\mathcal{X}$ be the set of open-ended forecasting questions; and $\mathcal{Y}$ the set of short textual answers. Let $x \in \mathcal{X}$ be a resolved forecasting question and $y^\star$ be the ground truth answer (as the question has already resolved. We ask the forecaster to respond with its best guess answer $y$, and the probability $q$ they assign to that being the true outcome. We evaluate this prediction tuple $<y, q>$ using the Brier score (Mucsányi et al., 2023) but adapt it to our setting. For a $K$-class outcome space $\mathcal{Y}$ with reported distribution $q$ and true class $y^*$, the (multi-class) Brier score is

$$S(q, k) = -\sum_{y \in \mathcal{Y}} (q_y - k_y)^2 = -(q_{y^*} - 1)^2 - \sum_{y \neq y^*} q_y^2,$$

where $k$ is one-hot encoding with $k_{y^*} = 1$. In our open-ended setting, $\mathcal{Y}$ is not predefined but rather its instances are provided by the forecaster. For simplicity, we elicit only a **single guess** $y$ with probability $q \in [0, 1]$ and assume the forecaster's probability is 0 for all other (semantically different) answers $y' \neq y$.[1] Applying the multi-class brier scoring rule in such a case induces a simplified score:

$$S(q, y, y^*) = \begin{cases} -(q - 1)^2 - 0 = -1 + 2q - q^2, & \text{if } y \equiv y^*, \\ -(0 - 1)^2 - q^2 = -1 - q^2, & \text{if } y \neq y^*. \end{cases}$$

Dropping the constant $-1$ yields

$$S'(q, y, y^*) = \begin{cases} 1 - (q - 1)^2, & \text{if } y \equiv y^*, \\ -q^2, & \text{if } y \neq y^*, \end{cases}$$

which shifts the range from $[-2, 0]$ to $[-1, 1]$ while providing a more natural interpretation: predicting $q = 0$ gives a baseline 0 regardless of $y$; correct answers receive positive scores, incorrect answers negative scores; and magnitude scales quadratically with confidence. We report $S'$ (visualized in Figure 8) as the *Brier score* in this paper.

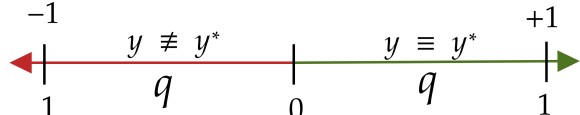

*Figure 8.* Illustration of the brier score when adapted to free-form response with answer $y$ and probability $q$.

Recent work by Damani et al. (2025) shows that this metric is a proper scoring rule, incentivizing both high accuracy and truthful reporting of probability on the answer that seems most likely. However, note that what we call as brier score here is distinct from the brier score considered by Damani et al. (2025). Their brier score is the one traditionally used for evaluating binary outcomes while ours is for free-form responses. Yet, we can show that our brier score is *same* as the **training reward** considered by them.

---

[1]This is technically incorrect to assume as the forecaster may have non-zero probability for guesses other than $y$. Ideally the forecaster should report all its guesses which have non-zero probability (with the multi-class brier scoring rule still being applicable) but we leave exploring this direction for future work.

# B. Analyzing `OpenForesight` Dataset

**News Corpus Details.** We drew from a heterogeneous pool of news articles diversified by geography, time, and topic to ensure broad coverage. We collected a English-language articles from outlets including Forbes, Hindustan Times, The Irish Times, Deutsche Welle, and CNN. They were selected by first collecting a corpus of approximately 250,000 articles spanning from June 2023 to April 2025, encompassing major large-scale events across sports, geopolitics, local news, crime, entertainment, and the arts. Then, performing de-duplication and filtering for language, text availability, and valid dates, we retained approximately 248,000 articles for the question generation phase. Table 3 details the distribution across news outlets for our corpus.

*Table 3.* Breakdown of source news articles by news outlet.

| Source | Articles (%Total) |
| --- | --- |
| Forbes | 110,103 (44.3%) |
| The Hindustan Times | 80,000 (32.2%) |
| The Irish Times | 29,546 (11.9%) |
| Deutsche Welle (DW) | 21,317 (8.6%) |
| Cable News Net (CNN) | 7,355 (3.0%) |
| **Total** | **248,321 (100%)** |

## B.1. Analysis of Best Question Selection

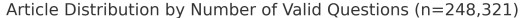

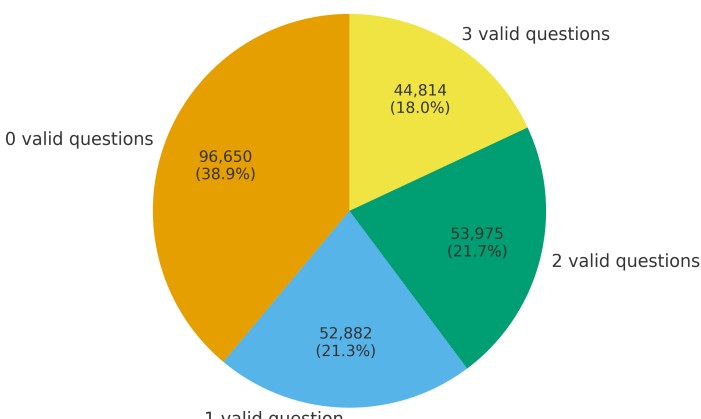

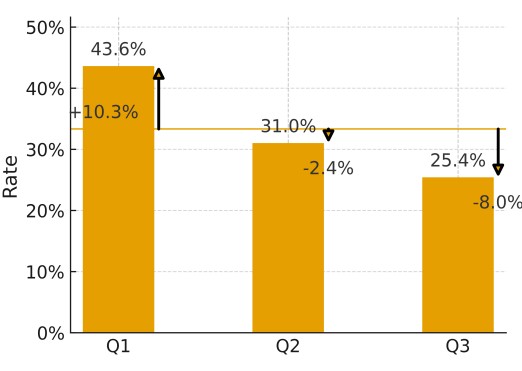

*Figure 9.* **Data Distribution of Questions in** `OpenForesight`. (Left) Distribution of number of questions selected after filtering from articles (Right) We show the number of questions generated, and the proportion of the first, second and third generate question being picked as the final "best question".

As Figure 9 (left) illustrates, the generation process yielded mixed results. Post-processing, 39% of source articles failed to produce any valid questions. Among the surviving articles, 21% yielded exactly one valid question, which we retained. For the 61% of articles producing multiple valid questions, we employed another LLM to identify the best candidate based on global relevance, specificity, and unambiguity. Our analysis of selected questions, shown in Table 9 (right) reveals a selection bias:

- Question 1: Selected 43.6% of the time (10.3% above random).

- Question 2: Selected 31% of the time (2.4% below random).

- Question 3: Selected 26.7% of the time (6.7% below random).

This trend suggests that initial generation attempts (question positioned earlier) frequently produce higher-quality results. Figure 5 provides qualitative examples of these generated questions.

## B.2. Distribution of Answer Types

Table 4 categorizes the answer types within the training data. Two categories dominate the dataset:

| | Name(s) | Location | Country | Title | Team name | Color | Organization | Currency | Brand name | Month |
|---|---|---|---|---|---|---|---|---|---|---|
| **Count** | 32,213 | 14,337 | 2,579 | 2,479 | 1,445 | 1,047 | 1,030 | 877 | 779 | 730 |
| **Share** | 44.8% | 20.0% | 3.6% | 3.5% | 2.0% | 1.5% | 1.4% | 1.2% | 1.1% | 1.0% |

*Table 4.* Top ten answer types of the questions in our curated dataset. These ten categories cover 80.1% of our training dataset.

- People and Places (65%): Names of individuals constitute nearly 45% of the answers, while locations account for 20%.

- Miscellaneous Entities (35%): The remainder consists of teams, countries, organizations, colors, and similar entities.

| Question | Background | Resolution (trigger & deadline) | Answer Type | Answer | Source |
|---|---|---|---|---|---|
| Host country of COP30 (Nov 2025)? | UNFCCC COP venue rotates among regions. | Host confirmed by UNFCCC/organizers; no later than COP30 start (Nov 2025). | string (country) | Brazil | DW: link |
| Release month of Marvel's *Fantastic Four* (2025)? | Reboot announced with lead cast; 2025 release slated. | Month confirmed by Marvel/Disney; by Dec 2025. | string (month) | July | Forbes: link |
| First state to require Ten Commandments in public classrooms (by 2025)? | Several U.S. states advance religion-in-school measures. | First state enacts requirement; by Dec 31, 2025. | string (state name) | Louisiana | Forbes: link |
| African host of G20 Summit (Nov 2025)? | G20 presidency rotates; South Africa presiding from Dec 2024. | G20/host government confirms location; by Nov 2025. | string (country) | South Africa | DW: link |
| Recipient of Lesotho–Botswana Transfer Scheme (by 2025)? | Regional pipeline to pump water from Lesotho via SA. | ORASECOM or governments confirm recipient; by 2025. | string (country name) | Botswana | DW: link |

*Table 5.* Five succinct forecasting questions spanning climate, entertainment, law, geopolitics, and infrastructure; selected for brevity and diverse sources (DW, Forbes). Each row lists the question (summarized here for conciseness), short background, resolution trigger with deadline, answer type, ground-truth answer, and citation.

## C. Qualitative Analysis of Final Answers

We manually annotated responses to 207 questions by both the initial Qwen3-8B thinking model and the trained `OpenForecaster`8B on the Guardian validation set. Using this set, we found that the agreement between the two models used for grading, Llama 4 Scout and Qwen3 4B is $\sim 97\%$, and we agree with their grading in over $\sim 95\%$ cases. This confirms the reliability of automatic answer matching based evaluation.

In Table 6, we analyze the domains (by news section) in which our trained model improves. We find significant improvements in the World, Australian, and US news sections, with no significant change for sports. This entails our model may not yet perform too well on sports-heavy prediction markets like Kalshi.

| Domain | $n$ | Before | After | $\Delta$ |
|---|---|---|---|---|
| world | 20 | 21.7 | 33.3 | +11.6 |
| australia-news | 15 | 35.6 | 42.2 | +6.7 |
| us-news | 21 | 41.3 | 44.4 | +3.2 |
| sport | 37 | 43.2 | 43.2 | +0.0 |
| football | 30 | 34.4 | 33.3 | -1.1 |

*Table 6.* Avg@3 by domain (n $\geq$ 10).

In Table 7, we analyze change in performance by question type, finding significant improvements on questions of the form "what", "which", and "who", while a slight regression in performance on location questions ("where").

| Question form | $n$ | Before | After | $\Delta$ |
|---|---|---|---|---|
| what | 25 | 14.7 | 29.3 | +14.7 |
| which | 98 | 45.2 | 51.4 | +6.1 |
| who | 60 | 27.8 | 33.9 | +6.1 |
| other | 10 | 40.0 | 43.3 | +3.3 |
| where | 14 | 47.6 | 45.2 | -2.4 |

*Table 7.* Avg@3 by question form (n $\geq$ 10).

Below, we present qualitative examples where our training improves and worsens predictions compared to the original model.

**Qualitative examples (improved; first sample)**

- **Q:** Who will be wearing the yellow jersey in the general classification at the end of stage eight of the 2025 Tour de France?
  **Truth:** Tadej Pogačar
  *Before:* Jonas Vingegaard (p=0.10)
  *After:* Tadej Pogačar (p=0.60)

- **Q:** Who will withhold a resolution from the U.S. House floor to force a vote on releasing the Epstein documents by July 25, 2025?
  **Truth:** Mike Johnson
  *Before:* Pam Bondi (p=0.30)
  *After:* Mike Johnson (p=0.60)

- **Q:** Which former Bank of England governor will be named in a Guardian piece criticizing 'moral hazards' for banks during the 2007–08 financial crisis?
  **Truth:** Mervyn King
  *Before:* Andrew Bailey (p=0.30)
  *After:* Mervyn King (p=0.40)

- **Q:** Which major tournament will the US women's national team focus on challenging for after the 2025 summer friendlies?

**Truth:** 2027 World Cup
*Before:* 2025 European Championship (p=0.95)
*After:* 2027 Women's World Cup (p=0.40)

**Qualitative examples (regressed; first sample)**

- **Q:** Which agency will drivers in Northern Ireland apply to for a replacement driving licence by 31 July 2025?
  **Truth:** DVA
  *Before:* DVLA (p=0.70)
  *After:* DVLA (p=0.20)

- **Q:** Where could Sweden's Euro 2025 journey conclude with a historic night if they continue to win?
  **Truth:** Basel
  *Before:* Basel (p=0.70)
  *After:* Zurich (p=0.40)

- **Q:** Who will be the Democratic Party's nominee for New York City mayor in the November 2025 general election?
  **Truth:** Zohran Mamdani
  *Before:* Zohran Mamdani (p=0.60)
  *After:* Andrew Cuomo (p=0.40)

- **Q:** Who will post the lowest first-round score among Rory McIlroy, Scottie Scheffler and Viktor Hovland at the 2025 Scottish Open?
  **Truth:** Viktor Hovland
  *Before:* Viktor Hovland (p=0.60)
  *After:* Scottie Scheffler (p=0.40)

### C.1. Test Set

For the test set, we first prepare an initial set of 1000 questions generated using `o4-mini-high` model as described in Section 6. We next performed additional filtering steps to retain high-quality future-facing questions:

1. We removed any potentially unanswerable questions (noise) by keeping only those which `grok-4.1-fast` could successfully answer majority of the times (run repeatedly, $n = 5$) with search tool access. This filtered out $15\%$ of the questions.

2. To address the issue of late reporting in news outlets, we again use `grok-4.1-fast` with search tool to find the **earliest resolution date** for a given question. This is important to prevent leakage from retrieving articles with the true answer. We retain only those questions with resolution date after May 2025. In Figure 10b, we report the number of questions per news source for whom the generated resolution date was within 1 month period of the date found by `grok-4.1-fast`. We notice an average of $70\%$ questions have resolution date even within the 1 month period.

Finally, we performed manual filtering of the remaining questions if the question did not satisfy validity criteria like: 1) having multiple answers 2) question actually resolves in the future (after September 2025) for which article reports the scheduled/planned place/event/etc. 3) question being irrelevant because it is too niche to a certain place or locality 4) question is about something which is already established (known). We provide news source specific filtering in Figure 10a, and the full guidelines we followed for manual filtering in the box below. All these filters led to a final test set of 302 questions.

---

**Manual Filtering Guidelines**

```
**Task:** You will be given (i) a news article and (ii) a question that is supposed
    to be answerable from that article. Your job is to decide whether to KEEP the
    question in the test set or REJECT it during manual filtering.

**How to review:**
1. Read the full question (Title, Background, Resolution Criteria, and the proposed
```

---

```
     Answer).
2. Apply the rejection criteria below. If **any** criterion triggers, REJECT the
   question. Refer to the article text if required.

   -----------------------------------------------------------
   **REJECT the question if it fails ANY of the following criteria:**

1) **Multiple possible correct answers (Ambiguous / not uniquely resolvable)**
   - Reject if the question could reasonably have more than one correct answer,
   even after reading the article.
   - Examples:
     - "Which company invested in ABC in July 2025?" when multiple companies are
     mentioned investing in ABC in July 2025.
     - "Who announced a new partnership?" when the article lists several
     partnerships.

2) **Resolves in the future (after September 2025)**
   - Reject if the question's true resolution depends on an event that happens
   **after 2025-09-30**, and the article only reports a
   *plan/schedule/announcement* rather than the event actually occurring.
   - Examples:
     - Article: "The ABC event is scheduled for Jan 2026 in PQR venue."
       Question: "Where will event ABC be held in Jan 2026?" -> Reject (the
     question resolves only after the event actually takes place as the venue can
     change last minute).

3) **Too niche / overly local / low general relevance**
   - Reject if the question is narrowly tied to a very specific locality,
   institution, or small community such that it is not broadly meaningful for a
   general test set.
   - Examples:
     - A minor municipal policy affecting only one small village.
     - A local event with no wider regional/national/global significance.

4) **Already established (Known / not a meaningful forecast target)**
   - Reject if the question is about something that is already settled or widely
   known at the time (i.e., not an uncertain outcome).
   - Examples:
     - "Who stepped down from the role of CEO of Company ABC in August 2025?"
   (requires knowing the current CEO which generally doesn't change frequently and
   is often stable over multiple years)
     - Long-established facts, definitions, or historical constants.
```

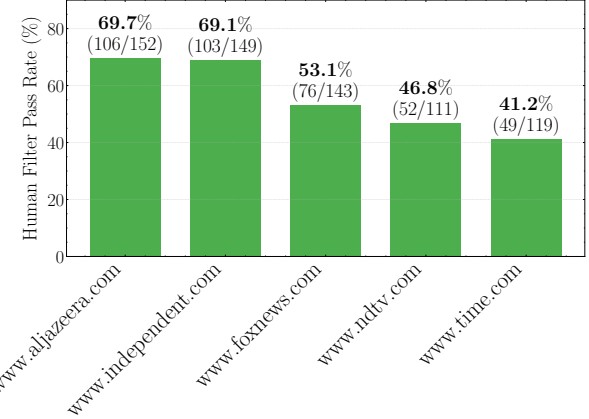

*(a)* Manual filter pass rate per news source.

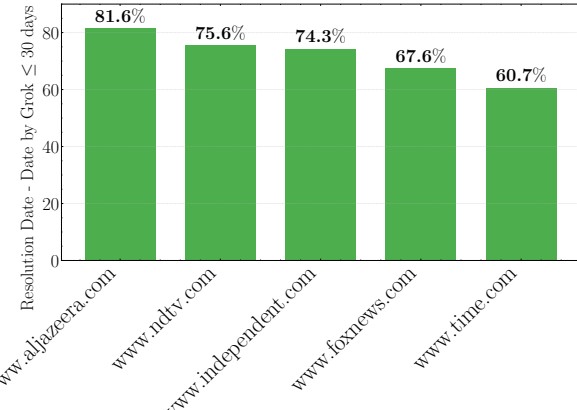

*(b)* Proportion of questions whose resolution date is within 1 month of the resolution date found by grok-4.1-fast.

*Figure 10.* Filtering pass rate of forecasting questions across news sources.

```
------------------------------------------------------------
**Decision format (for annotation):**
- If any criterion triggers, then **REJECT**.
- Otherwise, **KEEP**.

**Question:**
{questions_text}

<toggle> **Source Article:** {source_article} </toggle>
```

# D. Additional Results

## D.1. Ablation: Using Prediction Market Binary Data

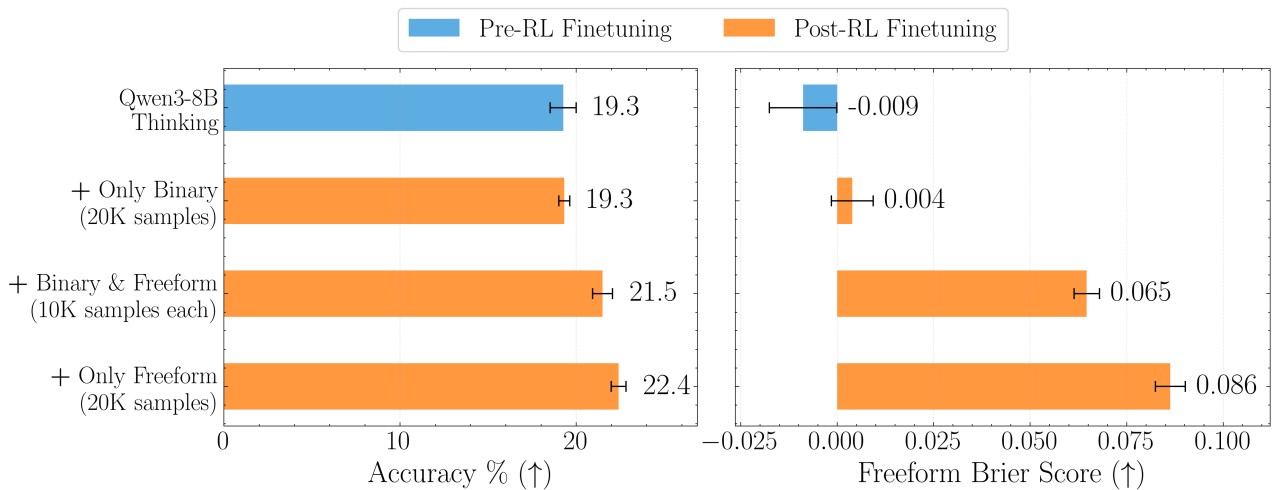

*(a)* Performance on our **Validation Set** composed of question from TheGuardian new source from July 2025.

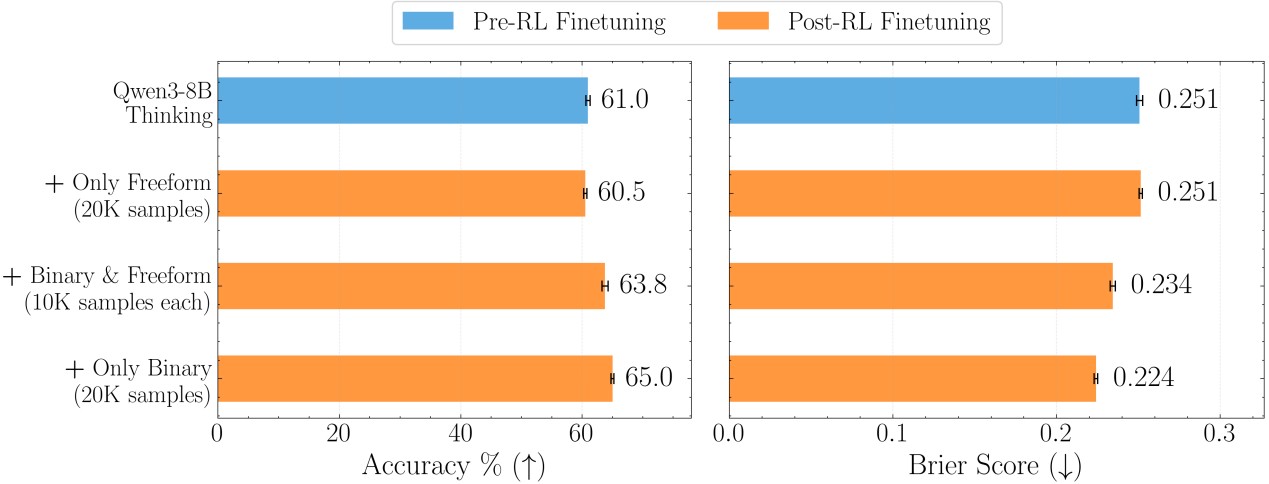

*(b)* Performance on **Metaculus binary** questions resolved in May–July 2025.

*Figure 11.* **Performance of different data ablations.** We evaluate performance after training on 3 different supervision signals: (i) only binary data (20K samples), (ii) only freeform data (20K samples), and (iii) both binary and freeform data (10K samples each) for data-matched comparison. (a) Accuracy and freeform Brier score of the initial and post-RL model on our Validation Set from July 2025. (b) Accuracy and binary Brier score of initial and post-RL model on volume-filtered binary questions resolved between May to July 2025 on Metaculus. *We find training on binary questions hurts performance on open-ended forecasting, but is necessary to retain performance on binary prediction market questions.*

We ablate supervision type with `Qwen3-8B` using three size-matched settings (Figure 11). For *binary-only*, we curate **20K** resolved markets from Manifold, volume-filtered to ensure engagement; because many markets resolve slowly, this set spans the past five years. For *free-form only*, we use **20K** pipeline-generated, usable questions from Forbes articles. For the *binary+free-form mix*, we take **10K** Manifold + **10K** Forbes questions to keep total examples constant. The goal is to isolate which *learning signal*—binary resolution vs. open-ended outcome specification—most effectively trains calibrated forecasters under identical compute and token budgets.

On the free-form test set (Fig. 11 Left), post-RL performance improves most with *free-form only* supervision (Accuracy 19.3% → 22.4%; Free-form Brier −0.009 → 0.086). Mixing binary and free-form also helps (Brier 0.065), whereas *binary-only* yields minimal gains on free-form evaluation (Brier 0.004). On Metaculus (binary) (Fig. 11 Right), both

*binary-only* and the *mixed* setting improve accuracy and Brier, with the *binary+free-form* mix offering the best overall trade-off across testing formats. Our gains by training on binary-only format are consistent with prior work by Turtel et al. (2025b;a). However, we do not arrive at a single unanimous recipe: free-form data is essential for open-ended forecasting, while combining formats appears Pareto-optimal across binary and free-form evaluations. Practically, it seems training on a *mixture* of question styles provides the most robust gains across tasks.

### D.2. Evaluation on Metaculus Questions

**Prediction Market Dataset.** We also source questions from the Metaculus prediction market from May - Nov. 2025. We filter questions which are either meta-prediction questions (prediction about some other prediction question on Metaculus), stock price prediction or below a certain trading volume leaving 449 high-interest questions. Each question is binary so a naive baseline gets 50% accuracy. We evaluate models on this benchmark by computing both accuracy and brier score.

**Results on Real Market Questions.** As shown in Figure 12, `OpenForecaster` 8B performs better than many larger models like DeepSeek-R1 and Llama-4-Maverick while GPT-OSS-120B is significantly better than rest of the models. **Note:** The brier score reported here is the *standard brier score* used in prediction markets ranging from $-1$ to $0$ (with -0.25 being the baseline for a constant prediction of $50\%$) and this is different from the brier score we have reported earlier for freeform questions.

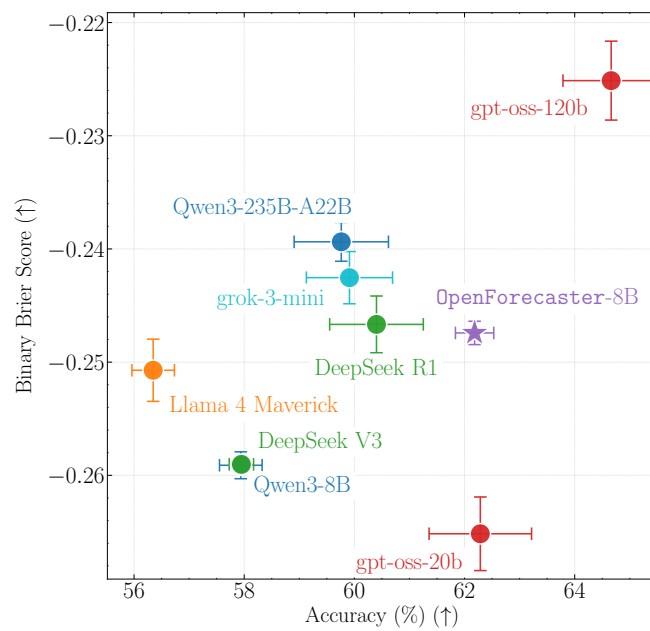

*Figure 12.* Performance of models on Metaculus questions from May 01 to Nov. 18 2025.

### D.3. Varying models and evaluation months

In Figure 13 we observe that while the first few article chunks that are retrieved lead to large improvements, at around five articles improvements plateau, both on our Qwen3-8B and also other large models like GPT-OSS-120B. Thus, unless otherwise specified, we use 5 articles for all evaluations and training in this work.

**Improvement on non-Qwen models.** Our training data `OpenForesight` can be used to improve models across different families. In Figure 14 we show improvements for Llama-3.1-8B-Instruct, Llama-3.2-3B-Instruct and Gemma-3-4B-Instruct. We see particularly large improvements in both accuracy and brier score for Llama due to both: poor initial performance, but also surprising amenability to RL training with our data as the final performance exceeds much larger models like Qwen3-235B-A22B and DeepSeek-v3.

**Results over time.** As our test set is derived from articles from May to August 2025, so we split the questions by resolution date to get monthly performance of the models. Breaking down by month, our test has 94 questions resolving in May, 85 in June, 76 resolving in July and 47 resolving in August. We have lower number of questions in later months due to our filtering strategy for addressing late reporting in news. We first generated roughly equal

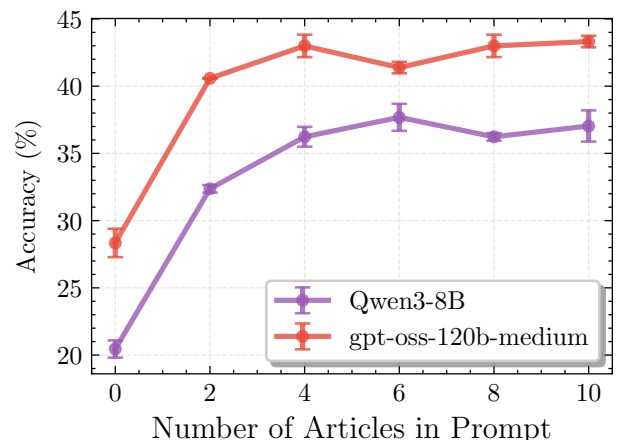

*Figure 13.* **Improvements from retrieval plateau at $\sim 5$ chunks.** We show the accuracy of both a large GPT-OSS-120B model and a small Qwen3-8B model which we finetune.

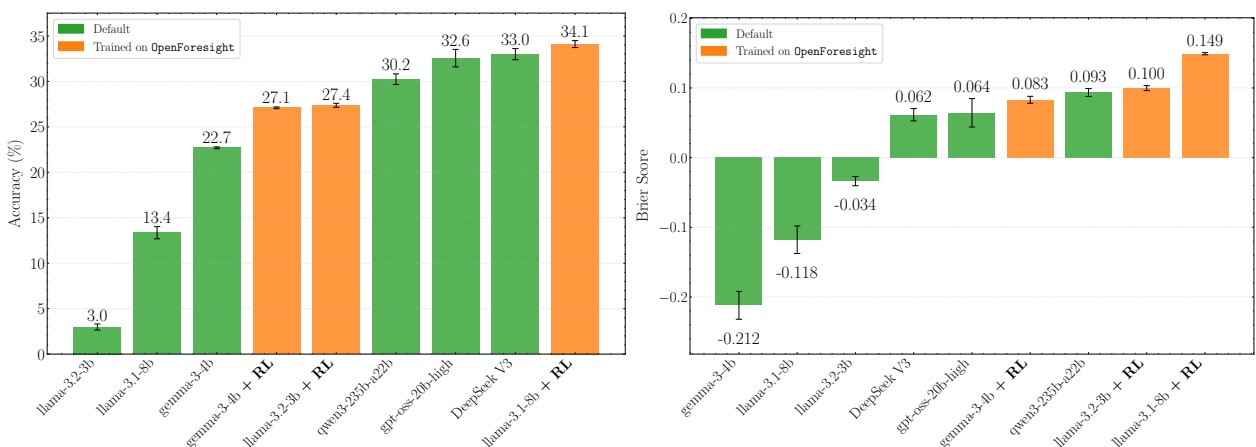

*Figure 14.* Performance of models from Llama and Gemma family our test set.

number of questions per month and post-hoc filtered the ones whose true resolution date (found using grok-4.1-fast with search) was before May'25.

For monthly performance, our hypothesis is that as we go further into the future, forecasting should become more difficult leading to lower performance. In Figure 16 and Figure 17, we find that the accuracy and brier score of the models indeed drops gradually month-by-month consistent with our hypothesis. We also find that our trained models are consistently better than the original versions and also better than all other models in Brier score.

### D.4. Ablation with Supervised Finetuning

Here we study what would be the benefit if we add a supervised finetuning (SFT) stage in our training process? While we start from the RL trained Qwen3 thinking models, they are far behind proprietary models as shown in Section 5. Several frontier model training reports (Guo et al., 2025) mention using an SFT stage as a warm start before RL. We choose Grok-3-Mini to generate forecasting reasoning traces for SFT, as it has high performance, low cost, and provides the full reasoning trace through the API. Specifically, we construct a dataset of 10,000 questions from *The Guardian* dated January–March 2025, beyond Grok-3-mini's reported knowledge cutoff of June 2024. Obtaining Grok-3-Mini's reasoning traces on this data costed 15 USD. For distillation, we randomly choose the number of articles to put in its prompt (0 to 10) so that the student model can reason with any number of articles.

Finally, we train this distilled checkpoint with GRPO using same data mixture, reward design and configurations as we used in training `OpenForecaster`-8B. We report the results for both just distillation (Qwen3-8B-sft) and RL on the SFT checkpoint (as Qwen3-8B-sft-rl) in Figure 15a on our curated test set from May to August 2025. The final model achieves higher accuracy and much better calibration albeit with a slightly lower brier score.

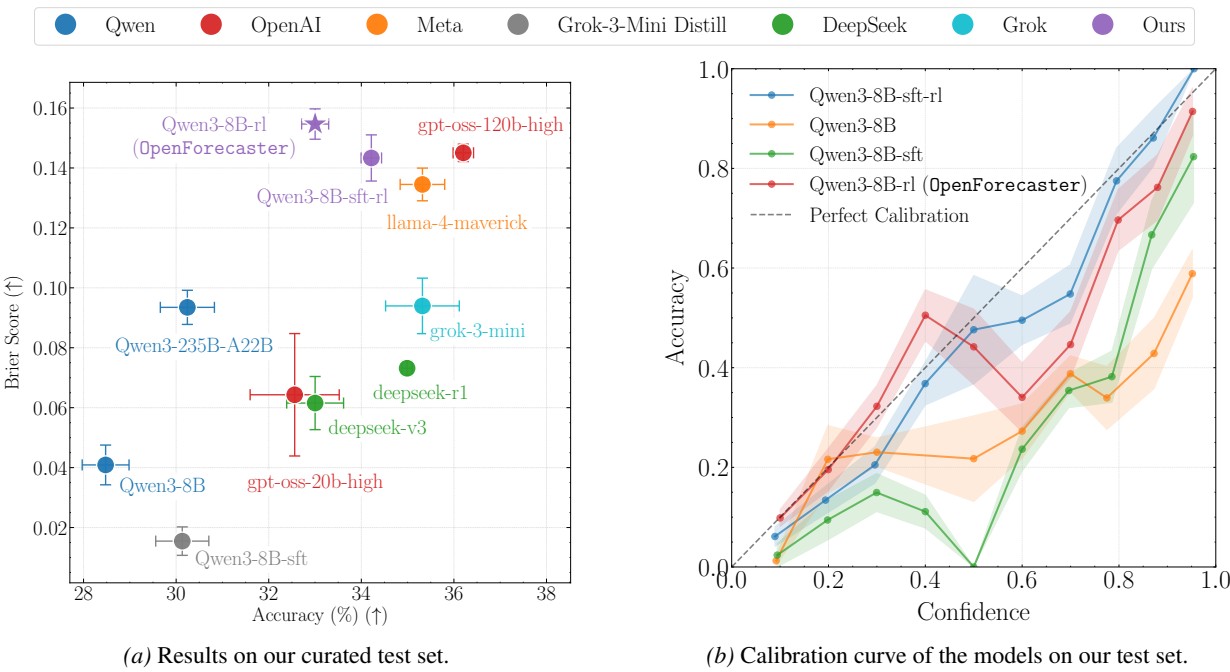

*(a)* Results on our curated test set.

*(b)* Calibration curve of the models on our test set.

*Figure 15.* RL training on `OpenForesight` improves the SFT models on accuracy. In particular, the calibration of Qwen-8B-sft-rl model (blue line) is near perfect.

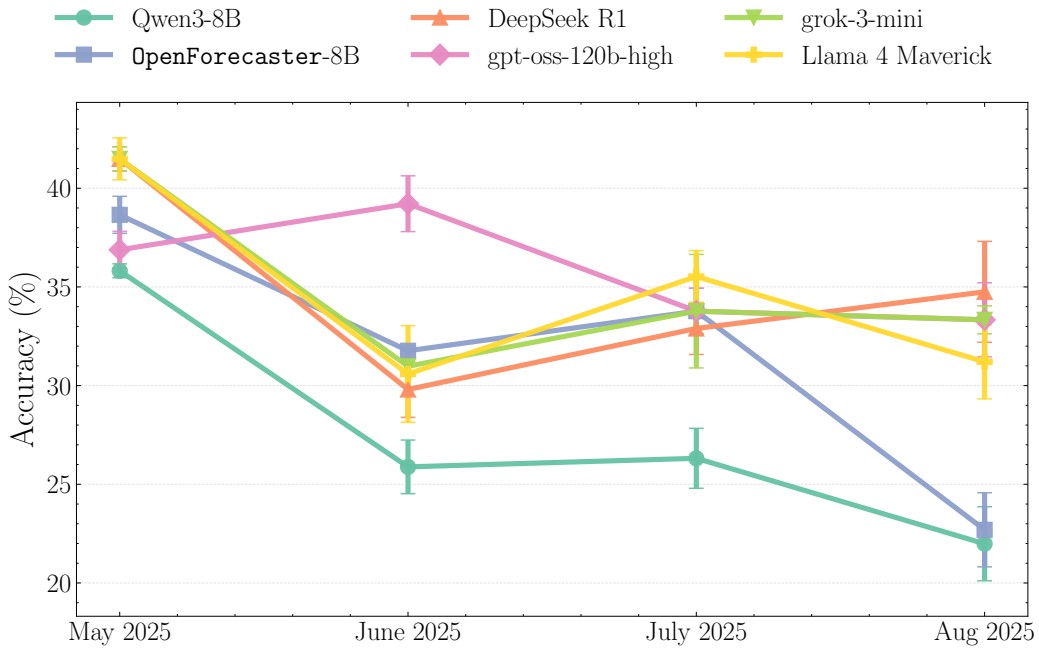

*Figure 16.* Monthly accuracy of the models on our test set.

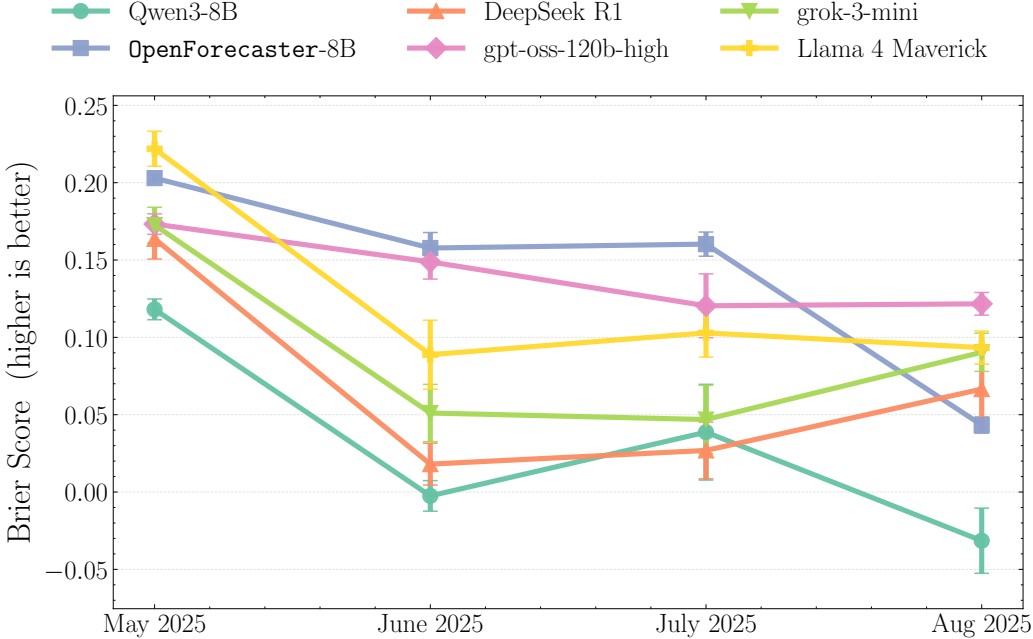

*Figure 17.* Monthly brier score of the models

# E. Qualitative Analysis of Reasoning Evolution During Training

We now provide a few qualitative examples of how the model reasoning evolves when training on OpenForesight. We ask the model itself to provide a summary of its reasoning (after the <think></think> tags), and show this summary below for the initial Qwen3 8B model, followed by training checkpoints.

## E.1. Example 1: Model stays incorrect but learns to hedge

**Question:** Which country will face a new 25% US tariff in a trade announcement by 30 July 2025?
**Background:** The US administration is evaluating other countries' trade barriers and is expected to announce new tariffs by the end of July 2025.
**Resolution Criteria:**

**Source of Truth**: Official tariff announcement issued by the White House or the Office of the United States Trade Representative.

**Resolution Date**: 30 July 2025, the date by which the 25% tariff announcement is expected.

**Accepted Answer Format**: The full name of the country as stated in the official announcement.

**Resolution answer:** India

*Checkpoint 0: score=-0.722, status=wrong*
<answer>South Korea</answer> <probability>0.85</probability> **Reasoning**: - **Article 2** (May 25, 2025) explicitly mentions a **25% tariff** on **Samsung** (South Korea) and other smartphone manufacturers. This directly ties the 25% tariff to **South Korea**. - While the EU is mentioned with a **50% tariff** in the same article, the **25% tariff** specifically targets South Korea's exports. - Other articles reference tariffs on China, the EU, or automakers, but none specify a **25% tariff** on a country beyond South Korea. - The resolution date (July 30, 2025) aligns with the timeline of the May 25 announcement, as no newer data is provided. **Confidence**: 85% (strong evidence from Article 2, though no post-May 25 updates confirm further announcements).

*Checkpoint 400: score=-0.360, status=wrong*
The retrieved articles indicate ongoing US tariff announcements by the Trump administration in 2025. Article 2 explicitly mentions a 25% tariff on South Korea for smartphone manufacturers like Samsung, while Article 4 references 25% tariffs on steel and aluminum imports from Canada and Mexico. Article 3 also notes a 25% tariff on foreign automakers, which could target countries like South Korea or others, but South Korea is specifically named in Article 2 for a 25% rate. Given the question's focus on a "new" 25% tariff announcement by July 30, 2025, and the patterns in the articles, South Korea appears as a likely candidate, as it has already faced such announcements and fits the criteria. However, the question implies a potential new announcement, and other countries like Canada or Mexico could also be possibilities based on Article 4, introducing uncertainty. My confidence is moderate due to the reliance on past announcements and the possibility of new developments by the resolution date. <answer>South Korea</answer> <probability>0.6</probability>

*Checkpoint 800: score=-0.090, status=wrong*
The retrieved articles indicate a pattern of US tariffs targeting major trade partners like China, India, the EU, Canada, Mexico, and South Korea. Article 4 specifically mentions a 25% tariff on steel and aluminum imports, which historically targets countries such as China and those in the EU. Article 2 highlights a 25% tariff on South Korean products (e.g., Samsung), while Article 1 and Article 3 reference ongoing tariffs on China and other nations. Given the US administration's focus on trade barriers, China is the most frequently mentioned target, making it a likely candidate for a new 25% tariff announcement by July 2025. However, the articles do not confirm a specific new announcement, and other countries like India are also potential targets based on Article 1, introducing uncertainty. Due to the lack of direct evidence for a July announcement and the possibility of shifts in US policy, my confidence is low. <answer>China</answer> <probability>0.3</probability>

## E.2. Example 2: Model goes from incorrect to correct

**Question:** Who will give Harvard one week to produce extensive records for its exchange visitor program review by July 23, 2025?
**Background:** The U.S. Secretary of State periodically issues directives to universities regarding their exchange visitor

program compliance.
**Resolution Criteria:**

**Source of Truth**: The official announcement or government statement naming the individual who issued the one-week deadline.

**Resolution Date**: When the directive is publicly attributed (expected by July 23, 2025).

**Accepted Answer Format**: Full name, for example "Jane Doe".

**Resolution answer:** Marco Rubio

*Checkpoint 0: score=-0.902, status=wrong*
The retrieved articles describe actions by the Trump administration's Department of Homeland Security, specifically Secretary Kristi Noem, toward Harvard. In the text, Noem sends Harvard a letter giving them 72 hours to provide information, not one week, so there is a mismatch between the question's "one week by July 23, 2025" and the articles' "72 hours." It seems that the July 23 date comes from the resolution criteria and isn't in the articles themselves, and that the 72-hour deadline might be part of a larger process leading up to that date, or that the question wording is slightly off. Despite this discrepancy, all the evidence points to Kristi Noem as the person issuing the directive to Harvard about providing records. Therefore, the answer should be Kristi Noem, and the confidence is high because all the articles point to her as the one who issued the directive to Harvard regarding the records.<answer>Kristi Noem</answer> <probability>0.95</probability>

*Checkpoint 400: score=-0.490, status=wrong*
The retrieved articles indicate that Kristi Noem, as the U.S. Department of Homeland Security (DHS) Secretary, has been directly involved in issuing deadlines related to Harvard's exchange visitor program compliance. For instance, in Article 2, she gave Harvard 72 hours (approximately 3 days) to comply with requests for records. The question specifies a one-week deadline by July 23, 2025, which may be a variation or continuation of similar directives from this administration. Given the pattern in the articles, it is reasonable to infer that Noem or a similar high-level official from the Trump administration is likely to issue such a directive. However, there is uncertainty due to the slight discrepancy between 72 hours and one week, and potential changes in personnel or administration by July 2025. Final Answer: <answer> Kristi Noem </answer> <probability> 0.7 </probability>

*Checkpoint 800: score=0.910, status=correct*
The retrieved articles primarily discuss the U.S. Department of Homeland Security (DHS) Secretary, Kristi Noem, in the context of Harvard's visa program issues, such as scrutiny and potential termination due to non-compliance. However, the question specifically pertains to the U.S. Secretary of State issuing a directive, as outlined in the question background. Based on the Trump administration's context in the articles, Marco Rubio was the Secretary of State in 2024 and likely remained in that role through at least 2025, as no other name is explicitly mentioned for that position. This is inferred from the articles' references to Trump administration actions, though the articles confuse DHS Secretary Noem with Secretary of State responsibilities. Given the resolution criteria requiring the official announcement, I am considering the most probable candidate from historical context, but the articles' inconsistencies lower my confidence. Final Answer: <answer>Marco Rubio</answer> <probability>0.7</probability>

### E.3. Example 3: Model goes from correct to incorrect, but interestingly reasons about brier

**Question:** Which Australian state besides New South Wales will be affected by the forecast 'second surge' of rain and wind on Wednesday night?
**Background:** The Bureau of Meteorology has forecast a 'second surge' of rain and wind for Wednesday night, expected to impact coastal regions.
**Resolution Criteria:**

**Source of Truth**: The question will resolve based on the Bureau of Meteorology's severe weather warnings and regional impact statements for the second surge, as published on bom.gov.au.

**Resolution Date**: 2 July 2025, following the Wednesday night updates.

**Accepted Answer Format**: The name of the Australian state, in 1–2 words.

**Resolution answer:** Victoria

*Checkpoint 0: score=0.978, status=correct*
The question's background says the resolution is on 2 July 2025, following Wednesday night updates. So the answer should be based on the Bureau's forecast as of that time. But the available articles are up to February 2025. If the 'second surge' is part of the forecast mentioned in Article 2 (from January 2025), then the answer is Victoria. However, the user might have a different scenario. Alternatively, in Article 4, there's a mention of damaging winds in parts of Illawarra, Sydney, and Hunter coast, but again, that's part of NSW. So, based on the available information, the most likely answer is Victoria. But I need to make sure that the 'second surge' refers to the same event. The article mentions a second strong wind coming from the south-west, which would affect the south of the country, including Victoria. Therefore, the answer is Victoria. Confidence: The articles mention Victoria as part of the affected regions. However, since the exact 'second surge' isn't explicitly named in the articles, there's some uncertainty. But based on the context, Victoria is the most likely state besides NSW. </think> <answer>Victoria</answer> <probability>0.85</probability>

*Checkpoint 400: score=0.840, status=correct*
The provided articles indicate patterns of weather impacts from the Bureau of Meteorology, particularly in eastern Australia. Article 2 explicitly mentions a weather system affecting eastern New South Wales, with parts of Victoria and south-east Queensland also likely impacted, including strong winds and rain. This aligns with the question's reference to a 'second surge' of rain and wind on Wednesday night. Article 1 and Article 3 further support influences on Victoria and South Australia from similar weather fronts, but Victoria is the most consistently mentioned state besides New South Wales across multiple articles. Given the resolution criteria relying on BoM's warnings, I infer that Victoria is the most probable state based on these patterns. However, the articles do not directly reference the specific forecast for 2 July 2025, introducing uncertainty. Confidence: The articles don't directly match the question, so I'm not 100% sure. Maybe 0.6, since it's inferred from patterns. To maximize my score with Brier score: If I'm correct, higher p is better; if wrong, lower p is better. I need to be honest about my confidence. I think 0.6 is reasonable, as it's a guess based on the info. <answer>Victoria</answer> <probability>0.6</probability>

*Checkpoint 800: score=-0.360, status=wrong*
The retrieved articles provide information on weather patterns in Australia, particularly from 2023 and early 2025. Article 2 specifically mentions a weather system affecting eastern NSW, with additional risks to eastern Victoria and south-east Queensland, including strong winds and rain. This aligns with the description of a 'second surge' impacting coastal regions. Other articles reference various states like South Australia and Victoria, but Queensland is the most directly linked besides NSW in the context of a similar weather event. Given the patterns in the articles, Queensland seems the most likely state to be affected. However, the articles are from different dates and not explicitly about the 2 July 2025 event, introducing uncertainty. Based on the evidence, I am moderately confident in this guess. I need to maximize my score using Brier scoring. If I'm correct, higher p is better; if wrong, lower p is better. Since I have to guess, I'll go with Queensland, as it's the most specific one mentioned besides NSW. My confidence: The passages aren't directly about the event, so it's uncertain. Maybe 0.4 or 0.5? Let's say 0.6, because it's the most prominent besides NSW in the relevant passages. <answer>Queensland</answer> <probability>0.6</probability>

## F. Systematic Failure Modes in Model Reasoning

`OpenForecaster` 8B achieved an accuracy of 33% on our test set consisting of 302 questions. Through manual analysis of the failure cases, we identified three systematic failure modes in the model's reasoning process.

**1. Missing Information in Retrieved Articles**    The most common failure mode occurs when retrieved articles lack the specific information needed. The model acknowledges this but falls back to general knowledge or educated guesses rather than expressing high uncertainty.

**Example 1:** "By what date will the New York City Board of Elections certify the official results of the Democratic mayoral primary?"

- **Correct answer:** July 15

- **Model answer:** August 15 (probability: 0.2)

- **Model's reasoning:** "The provided news articles discuss the New York City Democratic mayoral primary, including polling data, candidate strategies, and the election date of June 24, 2025. However, they do not specify the certification process or timeline for the New York City Board of Elections. Based on general knowledge of past NYC elections, certification typically occurs 2-6 weeks after the primary to allow for vote counting, ballot challenges, and legal reviews. For instance, in the 2013 primary, certification was completed by July 2. Given the late summer 2025 resolution criteria, a reasonable estimate is mid-August, but without specific historical data from the articles, this is uncertain. I am guessing August 15, 2025, as it aligns with typical procedures, but my confidence is low due to the lack of direct evidence."

- **Analysis:** The model correctly identifies that articles lack the certification date but incorrectly infers from general knowledge. It acknowledges uncertainty (probability 0.2) yet provides a specific date. The inference from "2-6 weeks after June 24" to "mid-August" is flawed, as July 15 falls within this range.

**2. Over-reliance on General Knowledge**    When articles lack specific information, the model makes educated guesses based on general knowledge, historical patterns, or analogies rather than acknowledging the answer cannot be determined from the provided context.

**Example 1:** "What will be the official name of Donna D'Errico's interactive AI voice service by mid-July 2025?"

- **Correct answer:** Call Donna D.

- **Model answer:** Donna Voice (probability: 0.1)

- **Model's reasoning:** "The provided articles do not mention Donna D'Errico or her specific AI voice service, making it impossible to draw direct insights. However, patterns from the articles, such as the naming of AI voice services (e.g., Melania Trump's "Melania: The Audiobook"), suggest that such services often incorporate the individual's name followed by descriptors like "Voice" or "AI." Based on this analogy, I am guessing the name might be "Donna Voice" or something similar. This is a speculative prediction, as there's no concrete information, leading to low confidence."

- **Analysis:** The model creates a naming pattern from one unrelated example (Melania Trump's service) and applies it incorrectly. It recognizes the speculation (probability 0.1) but the actual name "Call Donna D." follows a different pattern entirely.

**Example 2:** "What name will the new Pope take upon his election on May 8, 2025?"

- **Correct answer:** Leo XIV

- **Model answer:** Pius (probability: 0.1)

- **Model's reasoning:** "The provided articles do not specify the name of the next Pope or any potential candidates' chosen names. They focus on Pope Francis's health, the conclave process, and recent appointments of cardinals. Historically, new Popes select names that are traditional, often drawing from saints or biblical figures, but without specific information from the articles or external knowledge, any prediction is highly uncertain. Based on patterns from past Popes (e.g., Francis, Benedict), I am guessing "Pius" as a common regnal name, but this is purely speculative and not supported by the articles. Therefore, my confidence in this answer is very low."

- **Analysis:** The model relies entirely on general knowledge about papal naming conventions rather than specific information. It correctly identifies high uncertainty (probability 0.1) though. The reasoning conflates "common regnal names" with what the next Pope will actually choose.

**3. Entity Confusion: Selecting Wrong Person/Place/Organization**  When multiple entities are mentioned, the model frequently selects the most prominent or frequently mentioned one rather than the correct answer, even when both are present in the context.

**Example 1:** "Who will Chelsea face in the round of 16 of the 2025 FIFA Club World Cup?"

- **Correct answer:** Benfica

- **Model answer:** Bayern Munich (probability: 0.4)

- **Model's reasoning:** "The provided articles detail the groups for the 2025 FIFA Club World Cup, with Group C consisting of Bayern Munich, Benfica, Boca Juniors, and Auckland City, and Group D including Chelsea, Flamengo, Esperance Sportive de Tunisie, and a replacement (possibly Los Angeles FC or Club America based on Article 1). The question specifies that the round of 16 matchup is between the runner-up of Group D and the winner of Group C. Without the actual group stage results, I must make an educated guess based on team strengths. Bayern Munich is a strong candidate for the winner of Group C, as they are the defending champions and one of the top teams. Chelsea, in Group D, could plausibly be the runner-up, given their history, but this is uncertain. The replacement team adds further uncertainty, as it could affect group dynamics."

- **Analysis:** The model selects the more prominent team (Bayern Munich) from Group C rather than correctly identifying which team would actually win the group. It uses team prominence ("defending champions," "top teams") as a proxy for group stage results, ignoring that the question requires specific match outcomes that aren't in the articles.

## G. Prompt Templates

### G.1. Question Creation Pipeline

---

**Stage 1 — Question Generation (Requires: self.num_questions_per_article > 1)**

```
**Task:** Based on the provided news article, generate
    {self.num_questions_per_article} high-quality, DIVERSE forecasting questions
    which have a short answer (1 - 3 words), using the XML format specified below.
Each forecasting question should be posed in a way to predict future events. Here,
    the predictor will have a knowledge cutoff before the article is published and
    no access to the article, so a forecasting question has to be posed about
    information explicitly stated in the article. The question should be stated in a
    forward-looking manner (towards the future).
The correct answer should be a specific, short text response. The answer should be
    a WELL DEFINED, SPECIFIC term which the answerer can come up with on its own,
    without access to the news article.

**Example Format**:
<q1>
<question_id>0</question_id>
<question_title>Who will win the Nobel Prize in Literature in 2016?</question_title>
<background>Question Start Date: 10th January 2016. The Nobel Prize in Literature
    is awarded annually by the Swedish Academy to authors for their outstanding
    contributions to literature.</background>
```

---

```
<resolution_criteria>
<ul>
    <li>
      Source of Truth: The question will resolve when the Swedish Academy
    publicly announces the official 2016 Nobel Prize in Literature
    laureate(s)---typically via a press release on NobelPrize.org (expected on or
    about October 13, 2016).
    </li>
    <li>
      Resolution Date: The resolution occurs on the calendar date when the
    2016 laureate(s) are formally named
      (typically mid-October 2016).
    </li>
    <li>
      Accepted Answer Format: The full name of the laureate exactly as given
    in the announcement should be provided. If more than one person shares the
    prize, all names must be listed in the same order as the official communiqu{\'e}.
    </li>
</ul>
</resolution_criteria>
<answer>Bob Dylan</answer>
<answer_type>String (Name)</answer_type>
</q1>
```

The question should follow the structured guidelines below.

### **Guidelines for Creating Short Answer Forecasting Questions**

**Title Question Guidelines**
- **Quality**: The question should be of HIGH QUALITY and hard to answer without
  access to the article. It should not be about any minute details in the article.
  THE QUESTION SHOULD BE SUCH THAT ITS ANSWER REVEALS A KEY PIECE OF INFORMATION,
  FROM THE ARTICLE, WHICH HAS MAXIMAL IMPACT.
- **Specific and Answerable**: The question to be created SHOULD BE FREE-FORM and
  have a unique, specific answer (a single word, or short phrase) without access
  to the article. The answer to the question should be definite, well-defined and
  NOT NUMERIC. IT SHOULD ALSO NOT BE UNCERTAIN like "above XYZ" OR A RANGE LIKE
  "between XYZ and ABC". Avoid creating binary questions (yes/no, either/or) or
  questions with a list of specific options (multiple choice).
- **Answerable based on article**: Each question must have a CLEAR AND DEFINITE
  answer based on information stated in the article. Given the question, the
  content of the article should be able to resolve the answer to the question
  INDISPUTABLY WITHOUT ANY AMBIGUITY OR UNCERTAINTY. THE ARTICLE SHOULD NOT STATE
  THAT THE ANSWER IS TENTATIVE OR AN ESTIMATE OR LIKELY. The answer SHOULD HAVE
  HAPPENED BY NOW.
- **Temporal Information**: The question should not be about recall of (past) facts
  or events known before the article publish date. Include any temporal
  information necessary to answer the question (like by which month, year, etc.)
  in the question. The question should always be posed in a forward-looking manner.
- **Direct and Precise**: Titles must be straightforward and unambiguous, avoiding
  vague terms. Use future tense when appropriate.
- **Resolution Criteria**: ALWAYS INCLUDE A BRIEF RESOLUTION CRITERIA in the
  question title. This is often the date by which the question will be resolved.
  For example, resolution dates such as "by {{month_name}}, {{year}}?" or "in
  {{month_name}}, {{year}}?". THE RESOLUTION DATE SHOULD BE BASED ON (AND FAITHFUL
  TO) THE CONTENT OR PUBLICATION DATE OF THE ARTICLE.
- **No references to article or future information**: DO NOT refer to the specific
  article, such as by saying "in the article". The forecaster does not have access
  to the article, its metadata or any information beyond the article publish date.
- **Question Types**: Focus on "Who", "What", "When", "Where" questions that have
  concrete answers.
- **Understandability**: The question title should have ALL the information to be
  understandable by a 10 year old. It should be independently understandable

  without the article.
- **Tense**. ALWAYS POSE THE QUESTION IN A FORWARD-LOOKING MANNER. THE QUESTION
  SHOULD BE IN FUTURE TENSE. Try to use phrases like "What will", "Who will",
  "When will", "Where will", "How much/many will" etc. It should appear as a
  forecasting question and not past prediction.

**Answer Guidelines**
- **Faithfulness to Article**: The answer should be based on information explicitly
  stated in the article, and not implications or your own knowledge. IT SHOULD BE
  STATED VERBATIM IN THE ARTICLE.
- **Non-Numeric**: The answer should not be a number or a percentage. It can be a
  word, phrase, date, location, etc BUT NOT MORE THAN 3 WORDS.
- **Definite** – Given the question and the article, the answer should be CLEAR,
  CONCRETE, CERTAIN AND DERIVABLE from the article. It should be short,
  WELL-DEFINED TERM and not uncertain or vague. It SHOULD NOT BE A RANGE like
  "between XYZ and ABC" or "above XYZ" or "below PQR".
- **Resolved** – The answer MUST be something that has already happened or is
  happening now. It should be resolved given today's date and not be something
  that will happen in the future.
- **Specificity**: The answer should be specific enough to be unambiguous. Avoid
  overly general answers.
- **Conciseness**: Keep answers short – typically 1-3 words, occasionally a short
  phrase if necessary.
- **Exactness**: For names, use the exact names mentioned (full name, if possible).
- **Uniqueness**: The answer should be unique and THE ONLY CORRECT ANSWER to the
  question.
- **No Ambiguity**: The answer should be indisputable and not be open to multiple
  interpretations. IT SHOULD BE PRECISE AND NOT A RANGE OR UNCERTAIN ESTIMATE.

**Background Guidelines**
- **Mention Question Opening Date**: ALWAYS INCLUDE THE START DATE OF THE QUESTION
  IN THE BACKGROUND. IT SHOULD BE AT LEAST A FEW DAYS (OR WEEKS IF THE QUESTION IS
  ABOUT A LONG-TERM EVENT) BEFORE THE ARTICLE'S PUBLISH DATE AND ALSO BEFORE THE
  RESOLUTION DATE OF THE QUESTION. CONSEQUENTLY, THE BACKGROUND SHOULD NOT CONTAIN
  ANY INFORMATION WHICH HAS HAPPENED AFTER THE START DATE OF THE QUESTION.
- **Necessary Context**: The answerer does not have access to the article, so
  include MINIMAL CONTEXT required to understand the question keeping in mind the
  question opening date. Do not give (extra) details of the event from the article
  as background. If required, EITHER pose the event as a hypothetical scenario as
  if it were to happen in the future OR describe it as happening (unfolding) in
  real time. Describe any unfamiliar terms or concepts in the question title.
- **SHOULD NOT HELP ANSWER**: WHILE PROVIDING THE CONTEXT, DO NOT REFER OR MENTION
  OR LEAK THE ACTUAL ANSWER. The background must not help answer the forecasting
  question. DO NOT INCLUDE ANY INFORMATION from the article or elsewhere that
  either directly or indirectly (even partially) reveals the answer.
- **No Additional Knowledge**: Do not add any knowledge beyond what is required to
  understand the question. Only include information necessary to understand the
  question and its context.
- **Tense**. ALWAYS POSE THE BACKGROUND INFORMATION IN CURRENT TENSE. Only provide
  minimal information which is known until the question opening date.

**Resolution Criteria**
- **Necessary Criteria**: State the EXACT conditions by which the outcome will be
  judged. Include the criteria which determines how the question will be resolved.
  state the conditions by which the outcome will be judged.
- **Date and Source of Resolution**: Always state the date and the source by which
  the question will be resolved. For example, resolution dates such as "by
  {{month_name}}, {{year}}?" or "in {{month_name}}, {{year}}?", and potential
  source(s) of resolution such as "based on {{news source}}", "reports from
  {{official name}}", etc. THE RESOLUTION DATE SHOULD BE CHOSEN THOUGHTFULLY AS
  THE ANSWER'S VALIDITY AND SOUNDNESS DEPENDS ON IT. THE RESOLUTION DATE SHOULD BE
  SUCH THAT THE ANSWER CAN BE RESOLVED DEFINITELY AND INDISPUTABLY FROM THE
  CONTENT OR PUBLICATION DATE OF THE ARTICLE. IT SHOULD MENTION BY WHEN IS THE

```
    OUTCOME OF THE QUESTION EXPECTED TO HAPPEN. HOWEVER, IT SHOULD NOT LEAK OR
    MENTION ANYTHING ABOUT THE ARTICLE.
- **Details**: Be as detailed as possible in creating the resolution criteria for
    resolving the question as cleanly as possible. There should be no ambiguity in
    the resolution criteria.
- **Expectation and Format of Answer**: Based on the actual answer, the resolution
    criteria should state how precise the expected answer should be and in what
    format it should be. For example, if the actual answer is a date, the resolution
    criteria should specify how detailed the expected date should be -- only year,
    or both month and year, or day, month, and year all together. DO NOT GIVE THE
    ACTUAL DATE (ANSWER). If the actual answer is a percentage, then the criteria
    should state the expected answer should be a percentage. DO NOT GIVE THE ACTUAL
    PERCENTAGE.  If the actual answer is in certain unit, then the criteria should
    specify that. THE RESOLUTION CRITERIA SHOULD MAKE IT EXACTLY CLEAR AND PRECISE
    WHAT IS EXPECTED FROM THE ANSWERER AND IN WHAT FORMAT AND HOW IT WILL BE CHECKED
    LATER. IF GIVING AN EXAMPLE, IT SHOULD BE VERY GENERIC AND AS FAR AWAY FROM THE
    ACTUAL ANSWER AS POSSIBLE.
- **SHOULD NOT HELP ANSWER**: The resolution criteria must not directly help answer
    the forecasting question. DO NOT INCLUDE ANY INFORMATION from the article or
    elsewhere that either directly or indirectly (even partially) reveals the
    answer. DO NOT REFER OR MENTION OR LEAK THE ACTUAL ANSWER HERE.

**Answer Type Guidelines**
- **Expected Format**: The answer type should be either "numeric (XYZ)" if the
    answer is a number (of any kind) or "string (XYZ)" in all other cases. In
    numeric cases, XYZ should be the exact type of number expected. For example,
    "numeric (integer)", "numeric (decimal)", "numeric (percentage)", "numeric
    (whole number)", etc. In string cases, XYZ should broadly be the category of
    string expected. For example, "string (name)", "string (date)", "string
    (location)", etc. If the category is not clear, use "string (any)". HOWEVER,
    ALWAYS TRY TO CREATE QUESTIONS WHERE THE ANSWER CATEGORY IS CLEAR AND PRECISE.

**Question Quality Criteria**
- **Forecastable**: The question should be something that could reasonably be
    predicted or forecasted before the article's publication.
- **Towards the future**: THE QUESTION SHOULD BE POSED IN A FORWARD-LOOKING MANNER.
- **Interesting**: The question should be about a meaningful event or outcome, not
    trivial details.
- **Impactful**: The question should be such that if its answer is forecasted ahead
    of time, it should have significant (downstream) impact (relevant to high number
    of people).
- **Difficulty**: While the question should be hard to answer without access to the
    article, it should also not be unreasonably difficult.
- **Verifiable**: The answer should be something that can be EXACTLY verified from
    the article itself.
- **Time-bound**: Include clear timeframes or deadlines when relevant.
- **Free-form**: If possible, avoid creating binary questions (yes/no, either/or)
    or questions with a list of specific options (multiple choice).

Generate {self.num_questions_per_article} high-quality, DIVERSE short answer
    forecasting questions based on the provided article. Use the XML format with
    question_id value "0", "1", "2", etc. DO NOT INCLUDE ANY ANALYSIS, RANKING, OR
    ADDITIONAL COMMENTARY.

Article:
{source_article}

**Required Output Format**:
<q1>
<question_id>0</question_id>
<question_title>[Question 1]</question_title>
<background>[Background 1]</background>
<resolution_criteria>[Resolution Criteria 1]</resolution_criteria>
```

```
<answer>[Answer 1]</answer>
<answer_type>[Answer Type 1]</answer_type>
</q1>
..
<q{self.num_questions_per_article}>
<question_id>{self.num_questions_per_article - 1}</question_id>
<question_title>[Question {self.num_questions_per_article}]</question_title>
<background>[Background {self.num_questions_per_article}]</background>
<resolution_criteria>[Resolution Criteria
    {self.num_questions_per_article}]</resolution_criteria>
<answer>[Answer {self.num_questions_per_article}]</answer>
<answer_type>[Answer Type {self.num_questions_per_article}]</answer_type>
</q{self.num_questions_per_article}>
```

### Stage 2 — Individual Validation

```
**Task:** You will be provided with a news article and a question WHOSE ANSWER IS
    SUPPOSED TO BE BASED ON THE ARTICLE. Your job is to validate whether the answer
    to the question is valid by being faithful to the article (content, title, or
    description).

GO THROUGH EACH SEGMENT OF THE QUESTION ONE BY ONE (TITLE, BACKGROUND, RESOLUTION
    CRITERIA, ANSWER) TO UNDERSTAND THE WHOLE QUESTION. THEN CHECK EACH OF THE
    FOLLOWING CRITERIA:

1. **Tense and Details**: FIRST CHECK WHETHER THE QUESTION IS NOT UNDER SPECIFIED
    OR STATED IN PAST TENSE. IT IS FINE IF THE QUESTION IS STATED IN CURRENT OR
    FUTURE TENSE.
2. **Definite resolution of the answer by the article**: CHECK WHETHER THE ANSWER
    TO THE QUESTION IS SOUND, CLEAR AND PRESENT IN OR CAN BE DERIVED FROM THE
    ARTICLE. THE ARTICLE SHOULD RESOLVE THE ANSWER DEFINITELY AND IN AN INDISPUTABLE
    MANNER (WITHOUT ANY AMBIGUITY). THIS IS THE MOST IMPORTANT CRITERIA.
3. **Well-defined Answer**: The answer to the question should be short (NOT MORE
    THAN 3 WORDS). IT SHOULD NOT BE A PHRASE AND SHOULD BE SOMETHING WHICH IS
    CONCRETE, SPECIFIC AND WELL-DEFINED.
4. **Non-Numeric**: THE *ANSWER TYPE* SHOULD NOT BE NUMERIC LIKE A PERCENTAGE,
    INTEGER, DECIMAL, OR A RANGE.
5. **Single Correct Answer**: ANALYZE WHETHER THE QUESTION CAN HAVE MULTIPLE
    OUTCOMES OR RIGHT ANSWERS. IF SO, THE QUESTION FAILS THIS CRITERIA. OTHERWISE,
    ENSURE THAT THE PROVIDED ANSWER IS THE SOLE CORRECT ANSWER TO THE QUESTION. IT
    SHOULD NOT BE THE CASE THAT THE QUESTION CAN HAVE MULTIPLE (DISTINCT) CORRECT
    ANSWERS.

If ALL the above criteria pass (question is stated as required, answer to the whole
    question is valid, well-defined, and it is the only correct answer to the
    question), ONLY THENreturn <answer>1</answer>. Otherwise, return
    <answer>0</answer>. ALWAYS END YOUR RESPONSE IN <answer> </answer> tags.

**Article:**
{source_article}

**Question:**
{questions_text}

**Output Format:**
<answer>0/1</answer>
```

### Stage 3 — Choose Best

**Task:** You will be provided with a list of questions (possibly with size 1).
   Your job is to choose the best question from the list based on the following
   criteria or end your response with "NO GOOD QUESTION" if none of the questions
   meet the criteria.

**Instructions:**
GO THROUGH EACH QUESTION ONE BY ONE AND ANALYZE IT FOR THE FOLLOWING:
1. **Valid for forecasting**: Check if the WHOLE QUESTION is stated in a
   forward-looking manner. FROM THE PERSPECTIVE OF THE START DATE TO THE RESOLUTION
   DATE MENTIONED IN THE QUESTION, CHECK IF IT IS A VALID FORECASTING QUESTION. IF
   THE TIME HORIZON (START DATE TO RESOLUTION DATE) IN THE QUESTION IS AT LEAST A
   SINGLE DAY, THEN THE QUESTION SHOULD BE CONSIDERED VALID FOR FORECASTING. Go
   through each segment of the question (question title, background, resolution
   criteria) and check if each of them is valid and forward-looking.
2. **Tense**: The question SHOULD NOT BE STATED IN PAST TENSE. If the question
   covers an event, it should not imply as if the outcome of the event has already
   happened or occurred.
3. **Single Correct Answer**: ANALYZE WHETHER THE QUESTION CAN HAVE MULTIPLE
   OUTCOMES OR RIGHT ANSWERS. IF SO, THE QUESTION FAILS THIS CRITERIA. OTHERWISE,
   ENSURE THAT THE PROVIDED ANSWER IS THE SOLE CORRECT ANSWER TO THE QUESTION. IT
   SHOULD NOT BE THE CASE THAT THE QUESTION CAN HAVE MULTIPLE (DISTINCT) CORRECT
   ANSWERS.
4. **Impact**: How many people will the outcome of the question be relevant or
   interesting to? Consider on the basis of significant downstream impact or
   enabling meaningful action.
5. **Not Binary/Multiple Choice**: Question SHOULD NOT BE BINARY (yes/no, either
   ABC or XYZ, etc.) OR MULTIPLE CHOICE (SELECT FROM A LIST OF OPTIONS). It should
   be free-form (string -- name, date, place, etc.) or numerical (number,
   percentage, etc.).
6. **Understandable**: THe question as a whole (title, background, resolution
   criteria) should have sufficient details to understand the premise of the
   question. Every detail should be crystal clear and the question should not be
   under or over specified.
7. **Definite Answer**: EXTRACT THE ACTUAL ANSWER TO THE QUESTION PROVIDED IN ITS
   <answer> </answer> TAG. The extracted answer should be short, definite,
   well-defined and not uncertain or vague. It SHOULD NOT BE A PHRASE OR A RANGE
   like "between XYZ and ABC" or "above XYZ" or "below PQR".

ANALYZE EACH QUESTION BASED ON THE ABOVE CRITERIA ONE BY ONE AND CHOOSE THE ONE
   WHICH PASSES ALL THE ABOVE CRITERIA. IF MULTIPLE QUESTIONS SATISFY THE CRITERIA,
   CHOOSE THE ONE WHICH WILL HAVE THE HIGHEST IMPACT (AFFECTS OR IS RELEVANT TO THE
   MOST NUMBER OF PEOPLE). IF NO QUESTION MEETS THE CRITERIA, RETURN "NO GOOD
   QUESTION FOUND". OTHERWISE, RETURN THE BEST QUESTION IN THE SAME FORMAT AS THE
   INPUT.

**Generated Questions:**
{questions_text}

**Output Format:**
<q1>
<question_id>0</question_id>
<question_title>[ORIGINAL Title of the best question]</question_title>
<background>[ORIGINAL Background of the best question]</background>
<resolution_criteria>
<ul>
    <li> Source of Truth: [ORIGINAL Source of Truth of the best question]
    </li>
    <li> Resolution Date: [ORIGINAL Date of the best question] </li>
    <li> Accepted Answer Format: [ORIGINAL Accepted Answer Format of the

```
    best question] </li>
</ul>
</resolution_criteria>
<answer>[ORIGINAL Answer of the best question]</answer>
<answer_type>[ORIGINAL Answer Type of the best question]</answer_type>
</q1>
```

## Stage 4 — Leakage Removal

**Task:** You will be provided with a forecasting question. Your job is to ANALYZE
    whether the question's answer has obviously leaked in the content of the
    question. The question will have multiple segments -- question title,
    background, resolution criteria. EXCEPT THE QUESTION TITLE, GO THROUGH EACH
    SEGMENT STEP BY STEP and check if any part DIRECTLY leaks the actual answer. If
    leakage is found, ONLY THEN rephrase the problematic parts appropriately to
    remove the answer while maintaining the question's integrity and focus. DO NOT
    CHANGE ANY PART OF THE QUESTION UNNECESSARILY.

USE THE SAME XML FORMAT IN YOUR RESPONSE AS IS IN THE INPUT.

**Generated Question:**
{questions_text}

**Instructions:**
1. **Keep the title unchanged**: DO NOT MAKE ANY CHANGE TO THE QUESTION TITLE.
2. **Keep the start date in the background unchanged**: DO NOT MAKE ANY CHANGE TO
    THE QUESTION'S START DATE IN THE BACKGROUND.
3. **Identify the answer**: First, extract the actual answer from the XML tags for
    the current question being processed.
4. **Identify Leakage**: Keeping the extracted answer in mind, check if the
    background, or resolution criteria (each of them -- source of truth, resolution
    date, accepted answer format) contain information that reveals the answer.
5. **Types of leakage which can be ignored**: The following types of leakage are
    fine and don't need to be rephrased:
    – If the outcome (actual answer) of the question is binary (yes/no, either ABC
    or XYZ, etc.), then NO NEED TO CHANGE ANYTHING ANYWHERE.
    – If the resolution criteria is based on a list of specific options, then NO
    NEED TO CHANGE ANYTHING IN ANY SEGMENT (BACKGROUND, RESOLUTION CRITERIA, etc.).
    For example, if the accepted answer format states "answer must be either .." OR
    "answer must be one of the following terms..", then NO NEED TO CHANGE ANYTHING
    ANYWHERE.
6. **Types of Leakage to Check:** ONLY CONSIDER THE FOLLOWING KIND OF LEAKAGE:
    – DIRECT MENTIONS of the answer (either in word or number form) or part of the
    answer in the question/background/resolution
    – References to specific outcomes that ARE CLOSE TO (OR REVEAL)THE ACTUAL ANSWER
7. **Rephrase Strategy**: If leakage is found, rephrase the problematic part while:
    – Keeping the question's core intent
    – Maintaining forecasting nature
    – Preserving necessary context
    – Making the answer UNOBVIOUS by replacing with a FAKE ANSWER (FAKE NAME, DATE,
    NUMBER, PERCENTAGE, etc.) WHICH IS GENERIC AND NOT CLOSE TO THE ACTUAL ANSWER.
    – The rephrased part should not contain any information that is part of the
    actual answer. Neither should it indirectly hint or reveal the answer.
8. **Check Accepted Answer Format**: IF THERE IS ANY EXAMPLE MENTIONED IN ACCEPTED
    ANSWER FORMAT ("e.g..."), MAKE SURE THE EXAMPLE IS GENERIC AND AS FAR AWAY FROM
    THE ACTUAL ANSWER AS POSSIBLE. DO NOT INCLUDE AN EXAMPLE IF NOT MENTIONED
    ALREADY.
9. **Do not change the answer**: Do not change the actual answer to the question.
10. **Do not change the answer_type**: DO NOT MAKE ANY CHANGE TO the answer_type.
11. **Each segment should be checked independently**: Go through each segment of
    the whole question one by one. Everything from the title of the question to the
    background information to the resolution criteria should be checked

independently with reference to the answer of the question. In the resolution
    criteria, go through each <li> step by step. Do not change the other segments
    when rephrasing a problematic segment.
12. **Do not change anything unless leakage is found**: DO NOT UNNECESSARILY CHANGE
    ANY PART OF THE QUESTION UNLESS LEAKAGE IS FOUND.

IT IS ALSO POSSIBLE THAT MULTIPLE PARTS OF THE QUESTION HAVE LEAKAGE. YOU SHOULD
    CHECK EACH OF THEM INDEPENDENTLY AND ONLY IF LEAKAGE IS FOUND, REPHRASE THE
    PROBLEMATIC PARTS. DO NOT OVER-ANALYZE.

During your analysis, you should:
- Go through EACH SEGMENT OF THE QUESTION STEP BY STEP INDEPENDENTLY. First
    <background> and then inside <resolution_criteria>. Under the resolution
    criteria, go through the source of truth, resolution date, accepted answer
    format (each of them is a <li> tag) one by one. For each such segment, do the
    following:
    - Compare the content in the current segment with the actual answer. If ANY
    PART OF THE ANSWER is mentioned in the current segment, then consider that as a
    leakage UNLESS THE ACCEPTED ANSWER FORMAT IS BINARY (yes/no, either ABC or XYZ,
    etc.) OR A LIST OF SPECIFIC OPTIONS.
    - IF THE CURRENT SEGMENT IS BACKGROUND, DO NOT CHANGE THE QUESTION START DATE.
    - If the current segment is accepted answer format and there is a SPECIFIC
    EXAMPLE MENTIONED in it ("e.g. XYZ") which is close to the actual answer, then
    consider that as a leakage.
    - If leakage is found in the current segment, mention "Leakage found --
    {{reason for leakage}}". Form the segment with the problematic parts rephrased
    and mention it as "Replacement -- {{rephrased_text}}." THE REPHRASED TEXT SHOULD
    BE AS FAR AWAY FROM THE ACTUAL ANSWER AS POSSIBLE. It should now be present in
    the final output (instead of the original text).
    - Otherwise, mention "No leakage found". In your final output after you finish
    the analysis, return this segment UNCHANGED.
    - These outputs should be in the same format as the original input.
- Return the actual answer unchanged in the <answer> tag in your final output.
- Skip any other segments (question title, answer_type, etc.) in your analysis and
    output them unchanged (verbatim) in the final output.

Output your analysis step by step, and then end your response with the CORRECTED
    question in THE SAME XML FORMAT AS THE ORIGINAL.

**Output Format**:
{{ analysis }}

<q1>
<question_id>0</question_id>
<question_title>[UNCHANGED Question Title]</question_title>
<background>[Corrected Background]</background>
<resolution_criteria>
<ul>
    <li> [UNCHANGED Question Start Date] [Corrected Source of Truth] </li>
    <li> [UNCHANGED Resolution Date] </li>
    <li> [Corrected Accepted Answer Format] </li>
</ul>
</resolution_criteria>
<answer>[UNCHANGED Answer]</answer>
<answer_type>[UNCHANGED Answer Type]</answer_type>
</q1>

## G.2. Evaluation Prompts

---

**Model Evaluation Prompt (Uses retrieved news summaries)**

```
You will be asked a forecasting question (which might be from the past). You have
    to come up with the best guess for the final answer.
You will also be provided with a list of retrieved news articles summaries which
    you may refer to when coming up with your answer.

Please provide your reasoning before stating your final answer and also express how
    likely you think your answer is to be correct (your confidence in your answer).

Question Title: {question_title}

Question Background:{question_background}

Resolution Criteria: {resolution_criteria}

Expected Answer Type: {expected_answer_type}

Relevant passages from retrieved news articles:
{retrieved_news_articles_text}

Think step by step about the information provided, reason about uncertainty and put
    your final answer (in the format asked) in <answer> </answer>}tags.
You should also specify your confidence in your answer in <probability>
    </probability> tags.
The probability should be a number between 0 and 1.

You will be rewarded based on the probability (p) you assign to your answer.
Your answer will be evaluated using the BRIER SCORING RULE which is basically (- (1
    - p)\^{2}) if your answer is correct and (- 1 - p\^{2}) if your answer is
    incorrect.

For example, if p = 0.5}, and your answer is incorrect, then your score will be (-1
    - 0.5\^{}2) = (-1 - 0.25) = -1.25}
whereas if the answer was correct, then your score would be (- (1 - 0.5)\^{2}) = (-
    (0.5)\^{2}) = -0.25.

Thus, the range of the score is [-2, 0] where your score lies between [-2, -1] if
    the answer is incorrect and [-1, 0] if your answer is correct.

If your answer is correct, you will be REWARDED more if your probability is higher
    whereas if your answer is incorrect, you will be PENALIZED more if your
    probability is higher.
YOU HAVE TO MAXIMIZE YOUR SCORE.

Your final answer should be concise (NOT MORE THAN A FEW WORDS LONG) and your
    response SHOULD STRICTLY END with <answer> </answer> tags and <probability>
    </probability> tags.
```

---

**Example Prompt from Test Set (Without Retrieval)**

```
You will be asked a forecasting question (which might be from the past).
You have to come up with the best guess for the final answer.
Please provide your reasoning before stating your final answer and also express how
    likely you think your answer is to be correct (your confidence in your answer).

Question Title:
Which country in the Americas will report the highest number of chikungunya cases
    by July 15, 2025?
```

```
Question Background:
Public health agencies in the Americas are compiling chikungunya case counts for
    individual countries as the outbreak spreads in the region.

Resolution Criteria:
<ul>
  <li>Source of Truth: Cumulative case figures published by the Pan American
    Health Organization or the European Centre for Disease Prevention and
    Control.</li>
  <li>Resolution Date: July 15, 2025, when the mid-July regional report is
    issued.</li>
  <li>Accepted Answer Format: The name of the country in the Americas with
    the highest total reported chikungunya cases.</li>
</ul>

Expected Answer Type:
string (location)

Think step by step about the information provided, reason about uncertainty and put
    your final answer (in the format asked) in <answer> </answer> tags.
You should also specify your confidence in your answer in <probability>
    </probability> tags.
The probability should be a number between 0 and 1.

You will be rewarded based on the probability (p) you assign to your answer.
Your answer will be evaluated using the BRIER SCORING RULE which is basically (- (1
    - p)^2) if your answer is correct and (- 1 - p^2) if your answer is incorrect.

For example, if p = 0.5, and your answer is incorrect, then your score will be (-1
    - 0.5^2) = (-1 - 0.25) = -1.25
whereas if the answer was correct, then your score would be (- (1 - 0.5)^2) = (-
    (0.5)^2) = -0.25.

Thus, the range of the score is [-2, 0] where your score lies between [-2, -1] if
    the answer is incorrect and [-1, 0] if the answer is correct.

If your answer is correct, your will be REWARDED more if your probability is higher
    whereas if your answer is incorrect, your will be PENALIZED more if your
    probability is higher.
YOU HAVE TO MAXIMIZE YOUR SCORE.

Your final answer should be concise (NOT MORE THAN A FEW WORDS LONG) and your
    response SHOULD STRICTLY END with <answer> </answer> tags and <probability>
    </probability> tags.
```

## Example Prompt from Test Set (With Retrieval)

```
You will be asked a forecasting question (which might be from the past).
You have to come up with the best guess for the final answer.
You will also be provided with a list of retrieved news articles summaries which
    you may refer to when coming up with your answer.
Please provide your reasoning before stating your final answer and also express how
    likely you think your answer is to be correct (your confidence in your answer).

Question Title:
Which country in the Americas will report the highest number of chikungunya cases
    by July 15, 2025?

Question Background:
Public health agencies in the Americas are compiling chikungunya case counts for
```

```
     individual countries as the outbreak spreads in the region.

 Resolution Criteria:
 <ul>
   <li>Source of Truth: Cumulative case figures published by the Pan American
     Health Organization or the European Centre for Disease Prevention and
     Control.</li>
   <li>Resolution Date: July 15, 2025, when the mid-July regional report is
     issued.</li>
   <li>Accepted Answer Format: The name of the country in the Americas with
     the highest total reported chikungunya cases.</li>
 </ul>

 Expected Answer Type:
 string (location)

 Relevant passages from retrieved news articles:

 Article 1:
 Title: CDC warns US travellers of growing Dengue threat. Here's what you need to
     know
 Source: www.hindustantimes.com
 Article Publish Date: March 21, 2025
 Relevant Passage: CDC warns of rising dengue fever cases among U.S. travellers,
     reporting 3,484 cases in 2024, an 84% increase from last year. CDC cited a
     "record number" of cases reported among travellers in 2024, totalling 3,484
     cases. This marked an 84 percent surge compared to the previous year. "This
     trend is expected to continue with increased dengue activity in endemic areas in
     2025," the CDC stated in its warning. Dengue remains prevalent in certain
     regions of the States. The virus is being actively transmitted in U.S.
     territories, including Puerto Rico and the U.S. Virgin Islands, where outbreaks
     have been declared. Warmer temperatures during the spring and summer months
     create favourable conditions for the spread of the disease. Over the past five
     years, dengue cases have surged worldwide, with the Americas being notably
     affected. Data from the World Health Organization (WHO) reveals that in 2024
     alone, there were 7.6 million reported cases. Among these, 3.4 million were
     confirmed, over 16,000 were classified as severe, and more than 3,000 resulted
     in fatalities. Puerto Rico has been grappling with a sustained dengue outbreak
     since early 2024. The island surpassed the outbreak threshold in February of
     that year, leading to the declaration of a public health emergency in March
     2024, which remains in effect. Puerto Rico recorded 6,291 dengue cases in 2024,
     with more than half of the patients requiring hospitalization. Thirteen
     individuals lost their lives to the virus, according to the CDC data. Similarly,
     the U.S. Virgin Islands declared an outbreak in August 2024, which also remains
     ongoing. Health authorities reported 208 cases in 2024 and an additional 30
     cases in early 2025. The highest numbers of travel-related dengue infections in
     2024 were reported in Florida, California, and New York.

 Article 2:
 Title: Vaccine Against Chikungunya Approved By The FDA. Should You Get It?
 Source: www.forbes.com
 Article Publish Date: November 20, 2023
 Relevant Passage: efficient vector was Aedes aegypti; however, an interesting
     phenomenon occurred in 2005. There was a slight alteration of the virus genome,
     which allowed it to spread more efficiently with a more common mosquito, Aedes
     albopictus. That facilitated a massive pandemic in 2005 on La Reunion island and
     neighboring areas around the Indian Ocean. In 2013, chikungunya arrived in the
     Americas for the first time and it subsequently tore through the Caribbean
     islands. Is The United States At Risk? Yes. Before 2006, chikungunya rarely
     occurred in U.S. travelers. Then between 2006-2013, we had about 28 cases per
     year, but those cases were infected outside the U.S. The situation changed after
     the Caribbean outbreak and chikungunya arrived at our shores in 2014, with
     affected areas in Florida, the U.S. Virgin Islands and more severely in Puerto
```

Rico (over 30,000 suspected cases). The continental U.S. dodged a bullet, though, with fewer cases than feared. It probably helps that we have widespread air conditioning and window screens in our southern states, which reduces contact with mosquitoes. We remain vulnerable, though, since we still have the mosquito vectors, primarily in the central and southeast parts of the United States. How Can I Reduce My Risk? Minimizing risk focuses on avoiding mosquito bites when living in or visiting an area with active spread of chikungunya, including staying indoors in screened areas during the daytime, using bed nets, using insect repellant and wearing long, loose-fitting clothing. By avoiding Aedes mosquitoes, you reduce your risk of infection from chikungunya as well as dengue and zika viruses. Do I Need The Vaccine?

Article 3:
Title: First Chikungunya Vaccine Now FDA Approved – What To Know About The 'Emerging Global Health Threat'
Source: www.forbes.com
Article Publish Date: November 10, 2023
Relevant Passage: severe chikungunya-like adverse reactions following administration of Ixchiq." Big Number 5 million. There have been at least that many chikungunya cases reported globally over the past 15 years, the FDA said. The agency described the virus as an ''emerging global health threat" that ''has spread to new geographical areas causing a rise in global prevalence of the disease." What We Don't Know Health officials and agencies like the World Health Organization warn official infection counts are likely to significantly underestimate the true prevalence of chikungunya. Accurate diagnosis, disease surveillance and reporting can be tricky in some parts of the world on account of funding and capacity within healthcare systems and chikungunya is also ''easy to misdiagnose" on account of causing similar symptoms to other mosquito-borne illnesses like dengue and Zika. What To Watch For Valneva said the vaccine will initially address the ''potential needs" of some 60 million Americans who it says travel to countries where mosquito-borne diseases are endemic each year. This fits in well with its other shots for cholera and Japanese encephalitis aimed at travelers, the company said. Valneva said it will work towards commercializing the shot in the U.S. early next year and work towards securing a vote of approval endorsing the shot from the Centers for Disease Control and Prevention's vaccine advisory committee at the end of February. The FDA is an influential regulator and its go-ahead will likely speed Ixchiq's passage through other regulatory processes globally, particularly in areas where chikungunya is a more pressing concern. Key Background Chikungunya is regularly identified as an emerging threat to global health on account of the debilitating and prolonged disease it can cause. Chikungunya was first identified in Tanzania in 1952 and sporadic outbreaks were later recorded in parts of Africa and Asia. The virus has since spread globally and has been identified on all continents except Antarctica. The economic and social impact of the disease can be devastating -- the Coalition for Epidemic Preparedness Innovations estimates the cost to the Americas alone to be around $185 billion -- and the warming climate, a boon to the mosquitoes that spread the disease, is likely to widen areas at risk. While the virus is more commonly reported among travelers in the U.S. and parts of Europe, local transmission has been documented, suggesting future outbreaks may be possible or that the virus could gain a permanent foothold. Further Reading First Vaccine For Chikungunya

Article 4:
Title: Dengue fever cases rising in popular spring break locations, CDC alerts
Source: www.foxnews.com
Article Publish Date: March 24, 2025
Relevant Passage: is common in the Americas, Africa, the Middle East, Asia and the Pacific Islands, among other countries, according to the CDC. TRAVEL HOT SPOT SEEKS EMERGENCY DECLARATION OVER MASSIVE BUG INFESTATION In 2024, more than 13 million cases were reported in North, Central and South America, as well as in the Caribbean. Local transmission of these outbreaks was reported in California, Texas and Florida last year. Typical symptoms include aches and pains (in the eyes, muscles, joints, or bones), nausea, vomiting and rash -- usually

experienced within two weeks of being bitten. Most people experience symptoms
    for two to seven days before recovering. CLICK HERE TO SIGN UP FOR OUR HEALTH
    NEWSLETTER "It's typically a more mild illness, but can be severe, causing
    headaches, joint pain, fever, abdominal pain and even death," Dr. Mark Fischer,
    regional medical director of International SOS, a leading medical and security
    services company, previously told Fox News Digital. There is not currently any
    medication to treat dengue, according to the CDC. For more Health articles,
    visit www.foxnews.com/health Infected people are advised to rest, take
    acetaminophen for pain and fever, stay hydrated and see a doctor. There is a
    vaccine available for U.S. children between 9 and 16 years of age who have
    previously tested positive for dengue and are living in areas where the
    infection is common. CLICK HERE TO GET THE FOX NEWS APP

Article 5:
Title: Latin America: Key Themes To Watch In 2025
Source: seekingalpha.com
Article Publish Date: January 16, 2025
Relevant Passage: Latin America: Key Themes To Watch In 2025 Latin America's
    aggregate growth will slightly accelerate in 2025, but this overshadows slower
    growth across most countries. Read more here. Markit 3.25K Follower s ( 6min )
    Summary Latin America's aggregate growth will slightly accelerate in 2025, but
    this overshadows slower growth across most countries. The potential application
    of tariffs by the incoming US administration would negatively impact trade and
    weaken many of the region's currencies, while forced repatriation of illegal
    workers in the US implies a reduction of remittance flows. We expect lower price
    pressures based on Market Intelligence's global assumption that prices for
    agriculture-related commodities and oil prices will fall in 2025. In response to
    US President Donald Trump's proposals on tariffs and deportations, Mexico and
    most countries in Central America are likely to align with the requests from the
    new US administration. Here is how we see our key themes for 2025 shaping Latin
    America's operational and investment environment. Economic angst Latin America's
    aggregate growth will slightly accelerate in 2025, but this overshadows slower
    growth across most countries. We project only five This article was written by
    3.25K Follower s IHS Markit (Nasdaq: INFO) is a world leader in critical
    information, analytics and solutions for the major industries and markets that
    drive economies worldwide. The company delivers next-generation information,
    analytics and solutions to customers in business, finance and government,
    improving their operational efficiency and providing deep insights that lead to
    well-informed, confident decisions. IHS Markit has more than 50,000 key business
    and government customers, including 80 percent of the Fortune Global 500 and the
    world's leading financial institutions. Headquartered in London, IHS Markit is
    committed to sustainable, profitable growth. Comments Recommended For You
    Related Stocks SymbolLast Price% ChgEWZ--iShares MSCI Brazil ETFILF--iShares
    Latin America 40 ETFBRF--VanEck Brazil Small-Cap ETFFLN--First Trust Latin
    America AlphaDEX Fund ETFFBZ--First Trust Brazil AlphaDEX Fund ETF Related
    Analysis Trending Analysis Trending News

Think step by step about the information provided, reason about uncertainty and put
    your final answer (in the format asked) in <answer> </answer> tags.
You should also specify your confidence in your answer in <probability>
    </probability> tags.
The probability should be a number between 0 and 1.

You will be rewarded based on the probability (p) you assign to your answer.
Your answer will be evaluated using the BRIER SCORING RULE which is basically (- (1
    - p)^2) if your answer is correct and (- 1 - p^2) if your answer is incorrect.

For example, if p = 0.5, and your answer is incorrect, then your score will be (-1
    - 0.5^2) = (-1 - 0.25) = -1.25
whereas if the answer was correct, then your score would be (- (1 - p)^2) = (-
    (0.5)^2) = -0.25.

Thus, the range of the score is [-2, 0] where your score lies between [-2, -1] if

```
      the answer is incorrect and [-1, 0] if your answer is correct.

If your answer is correct, your will be REWARDED more if your probability is higher
      whereas if your answer is incorrect, your will be PENALIZED more if your
      probability is higher.
YOU HAVE TO MAXIMIZE YOUR SCORE.

Your final answer should be concise (NOT MORE THAN A FEW WORDS LONG) and your
      response SHOULD STRICTLY END with <answer> </answer> tags and <probability>
      </probability> tags.
```

# H. Details on Compute and Cost

To improve transparency around data and compute, we report approximate token counts, training steps, and GPU-hours for both SFT and RL. Our curated `OpenForesight` training set contains 52,183 samples. The average sample (just the question without any retrieval) has about 1000 characters and corresponds to roughly 400 tokens under the Qwen3 tokenizer for the question text, yielding approximately $2 \times 10^7$ prompt tokens in total.

For SFT, fine-tuning `Qwen3-8B` for 3 epochs took 5 hours on 8 H100 GPUs, corresponding to roughly 40 H100 GPU-hours. RL training is substantially more expensive: Our final run lasted for 5 epochs over the training set, resulting in about 1,300 optimization steps, for an estimated total of $\sim 1,000$ H100 GPU-hours. Including all ablations, we estimate we used $\sim 20,000$ H100 GPU-hours.

**Cost.** The overall dataset creation process costed us 3000$ with training set costing 2200$ (using DeepSeek-v3) while creating the test set costed 800$ (using o4-mini-high and grok-4.1-fast with search tool).

