# OpenReview forum: "Curating the Future: A Scalable Recipe for Training Open-Ended Forecasters"
_ICML.cc/2026/Conference — ICML 2026 regular_

### Official Review · Reviewer_6FnG · 2026-02-24

**Soundness:** 2
**Presentation:** 2
**Significance:** 3
**Originality:** 2
**Overall Recommendation:** 4
**Confidence:** 5

**Summary:**

The authors present OpenForesight, a curated dataset of approximately 50,000 open-ended questions about future events curated from news articles. They demonstrate that training Qwen3 thinking models with the proposed dataset and a loss function that combines accuracy and brier score loss improves brier score with results that are competitive to baseline LLMs.

**Compliance With Llm Reviewing Policy:**

Affirmed.

**Final Justification:**

The authors have addressed my concerns, so I have raised my score from a 3 to and 4.

**Key Questions For Authors:**

1. Are the model baselines in Fig. 6 trained from scratch or used zero-shot?

**Limitations:**

Yes

**Strengths And Weaknesses:**

Strengths
- The work presents a new dataset for open-ended questions with approximately 50,000 train samples.
- The authors open-source code, dataset, and models to enable reproducibility.
- Different train loss functions are ablated: Brier Score and Brier Score + Accuracy. The results show using both metrics achieves comparable accuracy as solely training with accuracy but results in improved Brier Score.
- In the curated dataset, the authors account for temporal leakage. They also leverage distinct news outlets for the test set to prevent journalistic style bias potentially observed in the train set from influencing the test set.
- The work compares performance of their pretrained model OpenForecaster-8B with several LLMs, including GPT, Grok, and DeepSeek models.

Main Weaknesses:

1. Limited novelty

- Open-ended prediction task as a novelty: The topic of "forecasting open-ended questions" isn’t clear from the introduction until page 2. However, it is still unclear how this functionality differs from general LLM tasks which address open-ended questions (not binary or categorical) from text prompts.

- Misleading contribution of methods to train models: "Guan et al. (2024);Wang et al. (2025) evaluate models on open-ended forecasts, but we go a step further by showing how to train models for this task." (Lines 119-121). Their approach to train models is presented as a novelty regarding related work yet the training procedure does not encompass new methods. The work uses preexisting model backbones, training methods (RL with GRPO), and loss functions (accuracy, brier score). If there are novelties regarding the training procedure which the authors refernce as a contribution, these should be explicitly discussed.

- Venue choice for dataset as main contribution: The main contribution appears to be the dataset. This paper would be better suited for a datasets and benchmarks track.

2. Reference to Forecasting: The use of the term 'forecasting' warrants clarification. In the quantitative forecasting literature, forecasting typically refers to the prediction of continuous numeric values. The task presented here focuses on the prediction of discrete events (Line 91) with unique answers that are short (1–3 words) and non-numeric, such as a name or location (Line 210). The authors should clarify how their use of 'forecasting' relates to or differs from the conventional definition. There may be a more precise term such as ‘temporal question answering’, which would align with Q&A tasks for LLMs while denoting the temporal aspect of the work.

3. Main evaluation: As the main contribution is the training dataset, I would expect the main evaluate to be training several models on this dataset to demonstrate benefits of proposed pretraining set compared to other pretraining strategies, however results on other models are deferred to the appendix: "Training on OpenForesight also improves models from other families like Llama and Gemma as we show in Appendix Section D.3." (lines 368-371).

Other/Suggestions:

1. Results should be discussed with quantitative support rather than qualitative language. Examples are below. The authors use percentage improvements is several areas, which is helpful (lines 380-384).
- Much larger - “Our specialized model, OpenForecaster-8B, matches much larger proprietary models on held-out testing, with our training improving both accuracy and calibration of predictions." (lines 26-31)
- By a large margin - “On FutureX, our model has the strongest accuracy by a large margin, even compared to much larger proprietary counterparts." (Lines 373-375).

2. The paper could benefit from revision to focus on dedicated related work, methods, and results sections. Several sections encompass overlapping information. For example, section 3 discusses related work and limitations of prior work, which should be discussed in the related work section. Another example is that both training details and results are discussed in section 5, and results are also discussed in section 6. Keeping information in dedicated sections would improve readability and prevent repetition.

3. Regarding the notation, model predictions are general denoted as $\hat{y}$ where $y$ is saved for the ground truth label

4. It is unclear whether Brier Score is generally used to train LLM models. There is one reference regarding this (“Beyond binary rewards: Training lms to reason about their uncertainty”) . Outlining related work (or lack there of) that use Brier Score and why it is important would be helpful.

6. Figure 4.: the results for accuracy and accuracy + brier score appear marginal (<1% improvement). Including confidence bands for this would help convey if the result improvement is significant or not.

---

> ### Author Rebuttal · Authors · 2026-03-30
>
> Thank you for your suggestions on writing, and we will incorporate them in the revised version. We hope to address your remaining concerns below.
>
> ### W1 - Limited Novelty
>
> > it is still unclear how this functionality differs from general LLM tasks which address open-ended questions
> >
>
> Our work differs from existing LLM open-ended questions e.g. chat datasets in specifically focusing on forecasting capabilities. This means the ability to predict questions about the outcomes of events that are in the future for the LLM, given its training / knowledge cutoff as described in the first 3 paragraphs of the Introduction. Past forecasting work had focused on binary and multiple choice questions, as discussed in Section 2 Related Work and Section 3 paragraph on “Motivation”. In the updated version, we will incorporate your suggestion on discussing what we mean by “open-ended” earlier.
>
> > Their approach to train models is presented as a novelty regarding related work yet the training procedure does not encompass new methods.
> >
>
> The novelty lies in how to scale training data for forecasting. We describe why this is important in the Introduction paragraph on “Scaling training data for forecasting”. Prior work depended on a much smaller set of resolved questions from prediction markets. We instead show how to obtain forecasting training questions from potentially any recent news article. Other novel insights in the specific context of forecasting training include using an offline corpus for retrieval to avoid leakage issues highlighted in past work, training on open-ended questions, and also finding that adding an accuracy bonus to the traditional Brier reward improves exploration. These are all described in Section 5.
>
> > The main contribution appears to be the dataset. This paper would be better suited for a datasets and benchmarks track.
> >
>
> The ICML call for papers includes “application-driven machine learning (innovative techniques, problems, and datasets that are of interest to the machine learning community” where we believe our scalable data curation recipe for forecasting fits well.
>
> ### W2
>
> > Reference to Forecasting: .. The authors should clarify how their use of 'forecasting' relates to or differs from the conventional definition.
> >
>
> mentioned in L143-147 where we start describing the motivation of the setting.
>
> ### W3
>
> > main evaluate to be training several models on this dataset
> >
>
> Note that Figure 1 is reported on Llama. As per your suggestion, we will include the Gemma and Llama results (currently in Appendix) to Fig. 6 in the main paper.
>
> ### Q1
>
> > Are the model baselines in Fig. 6 trained from scratch or used zero-shot?
> >
>
> The comparison is to larger, popular language models all released by major companies, and evaluated zero-shot.
>
> ### Other queries:
>
> > Figure 4.: the results for accuracy and accuracy + brier score appear marginal (<1% improvement)
> >
>
> In Figure 4, note how training for accuracy alone leads to a degradation in brier score (calibration), training for brier alone leads to a final accuracy of 25.5%, whereas training for both accuracy and brier score leads to high brier score (matching the brier only performance) and also higher accuracy ~27.5%). This is 2% absolute improvement on the accuracy axis, which is better with p < 0.01 based on bootstrap sampling.
>
> > It is unclear whether Brier Score is generally used to train LLM models… Outlining related work (or lack there of) that use Brier Score and why it is important would be helpful.
> >
>
> Brier score has been used extensively in past work in training LLM models **for forecasting**. For example, Turtel et al. (2025) and Halawi et al. (2024) which we reference in our work also evaluate  models primarily on brier score binary questions, whereas we extend this to open-ended answers.

---

> > ### Author Rebuttal · Reviewer_6FnG · 2026-03-31
> >
> > Thank you for your replies. I still have several unresolved concerns with the paper (explained below). As such, I will maintain my score.
> >
> > Thank you for clarifying that the ICML call includes novel datasets. I still believe the contributions referenced in the paper require revision, which was not addressed in the rebuttal. I believe the main contributions (generally) are: 1) a larger open-ended question dataset for training and evaluation, 2) a description of how to curate such a dataset, 3) evidence that training models with this dataset improves performance. Regarding methods on "how to scale training data for forecasting," if this is referring to data curation then this fits into my prior sentence. However, this work does not provide novel training paradigms, techniques, or loss functions as I mention in my review, which requires distinction in the paper. Also, the authors' claim that adding an accuracy bonus "improves exploration" is not supported. Exploration is never defined in the paper. Methods to analyze model exploration are non-trivial, and accuracy on held-out data does not necessarily reflect exploration.
> >
> > Regarding my comment on clarifying how their use of forecasting relates to or differs from the conventional definition: while the data characteristics are defined clearly (i.e., open-ended questions with temporal separation), my comment was referring to the forecasting task itself, where LLMs generally use classification rather than regression. Including notation regarding the forecasting task in this setting would be helpful, and help position it regarding models used for other data modalities, such as time series.
> >
> > Thank you for clarifying the percentage improvement. However, I still believe this 2% improvement is marginal. Explanation with regard to prior work improvements and how certain percentages translate to real-world impact would be helpful to better contextualize the value of such accuracy improvements.
> >
> > Regarding the bootstrapped results, bootstrap significance testing estimates variance over the test set but not over training. A p-value derived from a single trained model does not confirm that the result is stable; a different random seed may produce a different outcome. Results validated over multiple training runs or random seeds would be more convincing given the 2% difference, which is standard practice in forecasting literature (at least in time series forecasting).

---

> > > ### Author Response · Authors · 2026-04-01
> > >
> > > Thanks for the quick followup!
> > >
> > > We will revise the paper to acknowledge that we do not discuss novel training "paradigms" / "techniques" / "loss functions" in our paper and emphasize again that our contribution is on scaling training data. We will also clarify notationally how our work differs from time-series forecasting and uses LLMs as generative predictors rather than classification or regression.
> > >
> > > As for our discussion of adding an accuracy reward for improving "exploration", we will clarify that we meant exploration in a narrow sense of avoiding the model collapsing to not attempting hard questions at all during training. As discussed in L317-324 (R) we are specifically referring to the observation that when training only the brier score, improvements on accuracy plateau. This is because on hard questions the model learns the heuristic of providing predictions akin to "Unknown" with near-0 confidence. Consider the following illustrative example:
> > >
> > > > Question: “Which tech company will the US government buy a > 7% stake in by September 2025?”
> > >
> > > > Ground Truth: Intel
> > >
> > > > Response1: Answer: Unknown, Probability: 0.01 ; Brier Score = -1e-4; Our Reward = -1e-4
> > >
> > > > Response2: Answer: Intel, Probability: 0.01 ; Brier Score = 1e-4; Our Reward = 1 + 1e-4
> > >
> > > In this example, all predictions have brier score very close to 0 (brier score ranges from -1 to +1) since the probability assigned is 0.01, so it does not adequately reinforce reasoning chains that led to correct but uncertain guesses. As we reported, the behaviour of predicting "Unknown" with low confidence starts happening for ~40% questions when optimizing for brier score. Adding our accuracy bonus to the reward completely fixes this problem, with negligible "Unknown" attempts during training. On our validation set, we found our accuracy reward bonus helps the model do continued attempts eventually leading to a 2% accuracy improvement compared to brier-only training. We will add this illustrative example to our paper to clarify this point, and not use the word "exploration". Do you have any suggestions for this term that you'd like us to use instead, we were thinking "reducing unattempted predictions"?
> > >
> > > While we understand 2% is marginal usually, note that we are studying the really challenging task of forecasting. Here, 2% improvement can be really valuable depending upon the domain, for e.g. when predicting investment opportunities or political events. As per your request, we can compare this with prior work. Notably, Halawi et al. (2024), cited in our work, who also study training for forecasting had a <= 1% improvement in overall accuracy from their entire method (see Table 5, page 12 of https://arxiv.org/pdf/2402.18563), while for us this 2% is only for a single ablation. Thank you for bringing to our notice that we should contextualize our improvements in the paper.
> > >
> > > Regarding re-training, note that a single training run on OpenForesight takes ~1,000 H100 hours. This costs over 1,000$ per training run (for example, see https://www.genesiscloud.com/pricing which lists $1.6/per hour of single H100). Thus re-training models here is prohibitively costly for us, and not comparable to time-series forecasting. These costs are why reporting (especially ablation) results on a single training run with bootstrap sampling to estimate variance is a standard practice in LLM RL literature. That said, we will explicitly acknowledge this as a limitation in the final revision of our paper.
> > >
> > > Thanks for the engaged discussion and helping us improve our paper!

---

### Official Review · Reviewer_p7Zh · 2026-03-10

**Soundness:** 2
**Presentation:** 3
**Significance:** 3
**Originality:** 3
**Overall Recommendation:** 4
**Confidence:** 3

**Summary:**

This research training language models to make open-ended forecasts on real-world events. This work focus on the theme of scaling forecasting training data via automated curation of news. The CommonCrawl News corpus with static monthly snapshots are used to avoid leakage from online search. The specialized model OpenForecaster-8B matches or outperforms much larger proprietary models on Brier score with improved accuracy.

**Compliance With Llm Reviewing Policy:**

Affirmed.

**Final Justification:**

I think this is a pretty solid work, and my final score is 4.

**Key Questions For Authors:**

1. On the test set, how often does the top 5 retrieved context include the source article? What is accuracy with retrieval vs. without retrieval on this test set?

2. Can you provide a sensitivity analysis about train only on questions resolving in 2025 (or only in a narrow window before cutoff) to compare with current mix setting, and report validation/test impact?

**Limitations:**

See Weakness and Questions.

**Strengths And Weaknesses:**

## Strengths

1. The motivation is well stated, and the pipeline is fully automated and described with concrete stages, model roles, and prompts in the appendix.

2. The paper takes leakage seriously. This is a strong methodological stance and directly addresses known pitfalls.

3. The Accuracy and Brier reward is well motivated. The ablation supports the choice. Retrieval from the offline corpus gives large gains  and is implemented cautiously.

4. Evaluation covers a custom open-ended test set, FutureX, Metaculus, long-term consistency, and out-of-distribution calibration. The finding that forecasting training improves calibration on general benchmarks  is valuable and suggests the task encourages better uncertainty reporting.



## Weaknesses

1. Every training and test question is generated from a news article that reports the outcome. So the data generating process is from article published to question plus answer. For the model, the forecasting task can reduce to which document in the retrieval pool contains this question’s answer? The paper restricts retrieval to one month before resolution and uses different sources for test, but if the source article for a test question is in CCNews with a publish date within that window, retrieval could surface it and make the task trivial.

2. The base model’s knowledge cutoff is not officially reported, the paper assumes April 2025 and trains on events until then, testing on May August 2025. If the actual cutoff is earlier, then a large part of the future in training already in the past for humans at paper writing. That can inflate the perceived difficulty and the model’s forecasting signal. A short clarification of how cutoff was chosen would strengthen the temporal interpretation.

3. Answers are short within 1–3 words, mostly names and locations. The task is open-ended in the sense that the model must produce free form text, but the answer distribution is highly structured. The paper does not claim otherwise but the title open-ended may suggest broader scope.

4. The paper does not report confidence intervals, standard errors, or significance tests for the test set. Adding at least standard errors or a brief bootstrap would make comparative claims more sound.

---

> ### Author Rebuttal · Authors · 2026-03-30
>
> We are glad you like the soundness of our methodology in preventing leakage, and found the generalization from our forecasting training interesting.
>
> ### W1, Q1
>
> > …The paper restricts retrieval to one month before resolution and uses different sources for test, but if the source article for a test question is in CCNews with a publish date within that window, retrieval could surface it and make the task trivial.
> >
>
> Great question. Actually, we make sure this isn’t possible! The resolution date of a question generated from the article is set to $\min$(Grok 4.1 with web-search reported resolution date for the question, article date) as described in Appendix C.1. We then only retrieve articles from a reliably dated offline CCNews dump up to 1 month before this resolution date. Thus, the same article cannot show up in the retrieval, and there cannot be future leakage.
>
> Indeed, we now checked and confirm that it **never** shows up for any test set question. None of the retrieved passages are from the source article and only 0.7% (2 out of 302 questions) have partial overlap between retrieved passages and the source article (yet both of them don’t contain the answer in the retrieved passages).
>
> > What is accuracy with retrieval vs. without retrieval on this test set?
> >
>
> We do provide accuracy with and without retrieval on the validation set in Figure 5 (and the process of creating test set was similar to the validation set but with more quality checks). As per your request, we show below the accuracy on the test set ablating retrieval:
>
> | Model | Accuracy | Brier |
> | --- | --- | --- |
> | Qwen3-8B (no retrieval) | 21.63% ± 0.58% | -0.0420 ± 0.0062 |
> | Qwen3-8B (with retrieval) | 28.48% ± 0.51% | 0.0409 ± 0.0066 |
> | Qwen3-4B-Instruct-2507 (no retrieval) | 14.68% ± 0.77% | -0.1008 ± 0.0119 |
> | Qwen3-4B-Instruct-2507 (with retrieval) | 24.06% ± 0.80% | -0.0661 ± 0.0099 |
>
> As expected, retrieval significantly improves performance for both the models (+7-10% accuracy), and helps smaller model more since it has less parametric knowledge.
>
> ### W2, Q2
>
> > A short clarification of how cutoff was chosen would strengthen the temporal interpretation.
> >
>
> and
>
> > Can you provide a sensitivity analysis about train only on questions resolving in 2025 (or only in a narrow window before cutoff)
> >
>
> You are right that Qwen3-8B does not report a official knowledge cutoff. However, the model was released in 2025 after which its weights are frozen (and hashes tracked on Hugging Face) so it can only contain knowledge until April 2025. Thus, we test on events from May - August 2025 which the model definitely cannot have been trained on. It is true that some of the events in the training data are already known to the model. Since this is the training set, we are not particularly worried about this being true, as long as training on it leads to improvement on held-out test forecasting performance (which we ensure is truly in the future). In principle, our data curation pipeline can be run on articles after a model’s known cutoff / release date, and due to its scalability, generate a larger number of training forecasting questions per month. We will explore this in future work.
>
> As per your request, we trained Qwen3-8B on 10K questions resolving only in 2025 and found improvements but it did not match OpenForecaster-8B which benefits from being trained on 5x data, even if older.
>
> | Model  | Forecasting Training Data | Accuracy | Brier |
> | --- | --- | --- | --- |
> | Qwen3-8B | - | 28.48% ± 0.51% | 0.0409 ± 0.0066 |
> | OpenForecaster-8B | Full finetuning (50K) | 33.00% ± 0.29% | 0.1547 ± 0.0051 |
> | Qwen3-8B-2025 | 10K samples (from Jan to March 2025) | 31.46% ± 0.69% | 0.1211 ± 0.0081 |
>
> ### W3
>
> > Answers are short within 1–3 words, mostly names and locations…but the title open-ended may suggest broader scope.
> >
>
> We understand this concern and if you think thats better, are happy to omit the words open-ended from the title in our camera ready. We do think it accurately describes an important contribution of our work, in studying forecasting questions that are *substantially* *more* open-ended than prior work, which focuses on binary or multiple choice questions. We discuss this extensively in the “Motivation” paragraph in Section 3 and the Related Work Section. In particular, the major step we make is in not asking the model to choose from a predefined set of outcomes, but rather letting the model come up with plausible outcomes on its own just given the question. Even at 1-3 words, this raises the total possible outputs from 2-4 to vocabulary size^3, with LLM vocabularies being ~50,000 tokens!
>
> ### W4
>
> > The paper does not report confidence intervals, standard errors, or significance tests for the test set.
> >
>
> We actually do! Most figures in the main paper, except the data and retrieval ablations, include error bars (standard error over 3 trials). We’d be happy to extend our testing further if pointed to a particular experiment.

---

> > ### Author Rebuttal · Reviewer_p7Zh · 2026-04-01
> >
> > Thanks to the authors for their response. Most of my concerns have been addressed. I will keep my score unchanged.

---

> > > ### Author Response · Authors · 2026-04-01
> > >
> > > Thank you for your quick response. We noticed you have chosen '(b) - Partially resolved - I have follow up questions for the authors'.
> > >
> > > However, you did not share any follow-up questions but mention that most of your concerns have been resolved. Thus, we were wondering if you could please update your acknowledgement/score accordingly? Thanks!

---

### Official Review · Reviewer_FsPX · 2026-03-13

**Soundness:** 3
**Presentation:** 3
**Significance:** 3
**Originality:** 3
**Overall Recommendation:** 5
**Confidence:** 4

**Summary:**

This paper studies training models  for open-ended forecasing. Their main contributions are a new dataset and data generation pipeline, which lets forecasting questions be generated at scale, and leveraging this dataset to carry out RL training.  They demonstrate that  this training pipeline leads to increases in forecasting performance, including several ablations to highlight the imporant parts of their work.

**Compliance With Llm Reviewing Policy:**

Affirmed.

**Key Questions For Authors:**

- Do you expect training data contamination to affect your methodology?

**Limitations:**

yes

**Strengths And Weaknesses:**

Overall, I believe that training forecasters at scale is an important problem.  I found the methodology convincing, and believe all the claims are internally valid. Insofar as I have concerns, they are about the extent to which these results generalise to broader increases in performance around forecasting.

**Strengths**

* Data generation methodology: Previous works have explored how to generate forecasting questions from news articles, but have not achieved this at the scale present in this paper, and furthermore do not test this in an open-ended setting.
* Validation of the dataset through RL: Validating the dataset through RL training is an important contribution, and is novel within the literature.
* Ablations: I particularly liked the paper's ablations around the training methods, both on filtering and on using the Brier score.

**Weaknesses**

* Modest out-of-distribution generalisation: Overall improvements are highest on the authors' own held-out benchmark, which raises the question of whether the model is learning specific features of the benchmark, rather than general forecasting ability.
* Training data contamination*: When using this methodology, events being predicted may often have occured before the model's knowledge cutoff. Therefore, it seems plausible that models are being trained to better retrieve and answer facts which it already knows about, rather than more general forecasting skills - this may limit how strong gains in performance can be. Particularly for larger models, which may remember more of their dataset, I would be concerned about this.

Overall, I think the paper represents an important advance in the forecasting literature, and is further well-presented and technically sound. I overall recommend the paper to be accepted.

---

> ### Author Rebuttal · Authors · 2026-03-30
>
> We are glad you liked our scalable data creation recipe, and validation of it with RL experiments. We hope to address your remaining concerns below.
>
> ### W1
>
> > Overall improvements are highest on the authors' own held-out benchmark
> >
>
> We don’t think this is true. For example, Figure 6(b) shows large improvements on FutureX, an external benchmark, in both accuracy (>10% abs. improvement compared to 5% abs. improvement on our own test set) and brier score, even outperforming the much larger GPT OSS 120B. We also show improvements in the long-term forecasting consistency evaluation proposed in prior work in Table 2, and show calibration gains to completely out of distribution popular benchmarks like MMLU-Pro, GPQA and SimpleQA in Figure 7.
>
> ### W2 and Q1
>
> > Do you expect training data contamination to affect your methodology?
> >
>
> For our **test** set, we take extensive steps to eliminate any contamination from training data, more than any past forecasting study. Specifically, we do not rely on reported knowledge cutoffs of models. Instead, the models we use for training (Qwen3, Gemma3, Llama 3.1) were all released before May 1 2025, and our test set is filtered using Grok 4.1 + web-search removing questions which had resolved before May 1 as described in Appendix C.1. Finally, we use a reliably dated, offline dump of commoncrawl news for retrieval, and use our own search logic instead of web-search to avoid leakage of future information through retrieval, as described in L90-103 (”Ensuring we truly improve forecasting” paragraph in Introduction).
>
> For the **train** set, it is possible that events being predicted have occurred before model’s knowledge cutoff (which is unknown). However, we are not particularly worried about this, as long as training on it leads to improvement on held-out test forecasting performance (which we ensure is truly in the future).
>
> In Fig-6, we find it does lead to significant improvement on the test set showing that the model does learn forecasting skill/knowledge (even if trained partially on past events). To truly ensure models are trained on future questions, one would have to train only on events which happened after the release of model weights which we leave for future work. We encourage you to also check out the result we shared for Q1 of Reviewer `p7Zh` showing that training Qwen3-8B solely on 2025 data (10K questions resolving in 2025) also leads to significant improvements in Brier and Accuracy on the test set.

---

> > ### Author Rebuttal · Reviewer_FsPX · 2026-04-06
> >
> > Thank you! I will keep my current score.
> >
> > >For our test set, we take extensive steps to eliminate any contamination from training data, more than any past forecasting study. Specifically, we do not rely on reported knowledge cutoffs of models. Instead, the models we use for training (Qwen3, Gemma3, Llama 3.1) were all released before May 1 2025, and our test set is filtered using Grok 4.1 + web-search removing questions which had resolved before May 1 as described in Appendix C.1. Finally, we use a reliably dated, offline dump of commoncrawl news for retrieval, and use our own search logic instead of web-search to avoid leakage of future information through retrieval, as described in L90-103 (”Ensuring we truly improve forecasting” paragraph in Introduction).
> >
> > On my original read this wasn't completley clear, may be worth highlighting more promintently in the text.

---

### Official Review · Reviewer_F4y1 · 2026-03-13

**Soundness:** 3
**Presentation:** 3
**Significance:** 3
**Originality:** 2
**Overall Recommendation:** 5
**Confidence:** 3

**Summary:**

The paper studies the problem of training LLMs to become better at forecasting future events. The key innovations of the paper are as follows:
1. For this purpose, they develop an automated pipeline to extract questions from a news corpus (collected after LLMs cutoff date). Using this pipeline, they create a dataset of 52k questions for RL training.
2. They show that this dataset can be used to significantly enhance model performance on open-ended forecasting tasks.
3. They show that this training has nice side-effects, e.g., it improves calibration on GPQA and MMLU-Pro.

**Compliance With Llm Reviewing Policy:**

Affirmed.

**Final Justification:**

The paper is a simple and solid idea on a noteworthy application of LLMs. The execution of the paper is really good and authors addressed my main concern during the rebuttal effectively.

**Key Questions For Authors:**

1. Appendix E.3. Example 3 (line 1361). How did model develop awareness that it was being rewarded based on Brier score?
2. Figure 11 (and lines 371-372). Do authors have any intuition as to why it is important to train on binary questions for retaining performance on binary prediction market questions? What failure mode is observed if you don't include these in your training? Also I am not sure whether Figure 11 is using SFT or RL?
3. What is the distribution of the types of events in the dataset? Are they sport events, political events, financial events? The paper is motivated by the application of forecasting to social sciences domains, if so, can you explain what type of typical events are included in your dataset.
4. How correlated are mistakes / successes across models?
5. [Lines 433-436] Can you construct some sort of toy domain where optimal Bayesian updates are known and see how close LLM is to being a true Bayesian reasoner?

**Limitations:**

Yes

**Strengths And Weaknesses:**

### Strengths
1. The main strength of the paper is its simplicity. The core idea is quite simple and the paper does a good job of communicating this idea.
2. The paper includes number of interesting ablations that I found quite interesting.
3. I like the fact that for validation and test sets, the paper uses different news sources than for training set.
4. I found qualitative analysis of final answers in Appendix C quite interesting.

### Weaknesses
1. The paper's training is performing two functions that could result in improved forecasting performance: (a) you are doing reinforcement learning to train the LLM to reason better, (b) you are doing this on a specific task distribution. I would be interested in analysis that decouples gains in forecasting performance from (a) to (b). E.g., if we were to train the LLM on a diverse task distribution of reasoning tasks given the same compute budget, how much of the improved performance on forecasting benchmarks could be recovered?
2. The paper presents results primarily on one model of their own in Figure 6. Results on additional models are included in Figure 14 but this is not referenced in the main text.
3. Figure 6 does not include results about closed-source frontier models. I think this is likely logistically challenging (e.g, it is hard to know cut-off dates of these models) but if it is impossible, I would appreciate authors including some information as to why they don't include closed-source models in their evaluation.
4. The dataset does not have any difficulty markers, so, it is unclear how hard the questions are.
5. The test set is only a bit more than 300 questions (which is much smaller than training set of 52k questions). I would have ideally liked it to be bigger.

**Minor:**
- Figrure 4 does not say which model is being used here.
- Typo in line 355 (should be Figure 5 not Section 5).

---

> ### Author Rebuttal · Authors · 2026-03-31
>
> We are glad you liked our experiments and approach! Below we discuss how we incorporate your suggestions:
>
> ### W1
>
> To decouple gain from forecasting training with reasoning training, we use Llama-3.1-8B-Instruct which initially did not undergo reasoning RL. We compare training it on OpenForesight (as reported in Appendix D.3) with the performance of Llama-3.1-Nemotron-Nano-8B-v1 (https://arxiv.org/abs/2505.00949) which underwent post-training (SFT + RL) on a diverse corpus of 33M reasoning and chat samples, on **our** forecasting test set.
>
> | Model | Accuracy | Brier |
> | --- | --- | --- |
> | Llama-3.1-8B | 13.36 | -0.1179 |
> | Llama-3.1-Nemotron-Nano-8B | 16.67 | -0.1061 |
> | Llama-3.1-8b Forecasting RL | 34.11 | 0.1492 |
>
> We see that Nemotron improves only by 3% while our training improves it by 20% (abs) and the brier also improves significantly (from -0.1 to +0.15) despite OpenForesight being 1000x smaller than the data used by Nvidia to train Nemotron. This clearly shows a significant amount of improvement is coming from our targeted forecasting training.
>
> ### W2
>
> > The paper presents results primarily on one model of their own in Figure 6. Results on additional models are included in Figure 14 but this is not referenced in the main text.
> >
>
> In L368 (R) we do mention in passing that we trained and improved Gemma and Llama. We will highlight this more clearly in the revision.
>
> ### W3
>
> We did not include closed-source frontier models as their knowledge cutoffs are not transparent, so it is hard to make a clear comparison and only included `grok-3-mini` as it was released in Feb 2025 (thus, before our test set) which our model matched in performance.
>
> We now add GPT-5, Grok-4.1-Fast, and Gemini-3-Flash — released around July 2025, so potentially contaminated on our May–August 2025 test set. All were evaluated with `high` reasoning effort. While all frontier models achieve higher accuracy than OpenForecaster-8B, interestingly our model significantly outperforms Gemini-3-Flash on Brier.
>
> | Model | Accuracy | Brier |
> | --- | --- | --- |
> | gemini-3-flash-preview | 46.47% | 0.1094 |
> | gpt-5 | 45.03% | 0.2389 |
> | grok-4.1-fast | 40.73% | 0.1850 |
> | OpenForecaster-8B | 33.00% | 0.1547 |
>
> ### W4
>
> > The dataset does not have any difficulty markers
> >
>
> This is true. We do not know of a better way to quantify difficulty than the model accuracies themselves, which are reported.  We would be happy to incorporate suggestions.
>
> ### W5
>
> We now test on a new larger test set of 832 forecasting questions we created from AlJazeera articles between Sept - Dec 2025. We used GPT-5.2 to create questions and followed the same filtering strategy as used for the original test set described in Appendix C.1. Below are the results:
>
> | Model | Accuracy | Brier |
> | --- | --- | --- |
> | Qwen3-8B | 30.65 | 0.0465 |
> | OpenForecaster-8B | 35.34 | 0.1741 |
> | llama-3.1-8b-instruct | 13.78 | -0.1196 |
> | llama-3.1-8b-instruct + Our RL | 35.22 | 0.1799 |
>
> We find similar improvements as the original test even though the questions are more in the future, showing our results are robust.
>
> ### Q1
>
> > How did model develop awareness that it was being rewarded based on Brier score?
> >
>
> Appendix G.2 has the evaluation prompt which explicitly informs the model that it will be tested on the brier score, with the full formula.
>
> ### Q2
>
> We observed that when we don’t train on binary questions, performance on binary questions is lower than training on both open-ended and binary questions. We believe this is a generalization issue with GRPO training that is beyond the scope of this study to fix. Regarding your second question, Figure 11 is only using RL.
>
> ### Q3
>
> The distribution of events is based on the news sources they are generated from for this work. We perform the topic-wise analysis as you suggested and will include it in the revision:
>
> We found 21% questions are about politics, 18% about sports, 11% about conflicts/security, 11% about business/economy, 10% about culture/lifestyle, 10% about legal/justice, and the remaining are spread across topics like infrastructure, medicine, climate, technology etc.
>
> We think this is a healthy mix representing a range of topics people care about, as reported in the news.
>
> ### Q4
>
> > How correlated are mistakes / successes across models
> >
>
> We compute pairwise Pearson correlations on binary outcomes (correctness) on all models that were in Figure 6. We find **strong** correlation, ranging between 0.54 to 0.83 (mean 0.71), with 48.3% questions missed by all models, and 17.2% solved by all. OpenForecaster-8B (ours) is the least correlated with other models, ranging from r=0.54 to 0.63, suggesting our RL training data introduces independent signal.
>
> ### Q5
>
> > how close LLM is to being a true Bayesian reasoner?
> >
>
> This is outside the scope of our current work. Parallel work https://www.nature.com/articles/s41467-025-67998-6 might be of interest.
>
> Thanks for bringing up the two typos, we will fix them in the revision.

---

> > ### Author Rebuttal · Reviewer_F4y1 · 2026-04-01
> >
> > Thanks for the response. I found it to address most of my concerns. However, I think it would be useful to decouple paper’s motivation from potential applications to social sciences in the introduction. This motivation is not really explored further within work, and I think forecasting of world events is an important problem that stands on its own (which is what the paper focuses on).

---

> > > ### Author Response · Authors · 2026-04-06
> > >
> > > Thank you for your response! We are glad to have resolved most of your concerns and we will update the introduction in the revision.

---

### Decision · Program_Chairs · 2026-04-30

**Decision:**

Accept (regular)

**Comment:**

Reviewers agreed that forecasting is an important application area and that this paper contributes a thoughtful large-scale recipe for constructing forecasting data and training/evaluating open-ended forecasters. They were broadly positive about the anti-leakage precautions, the care taken with temporal separation and retrieval design, and the evidence that the resulting training improves both accuracy and calibration on forecasting tasks.

The remaining debate was not about a fundamental soundness problem, but about how to position the contribution. Some reviewers asked whether the gains should be understood primarily as evidence for forecasting-specific training or more generally as a result of large-scale reasoning-oriented training and careful data curation. One reviewer in particular remained concerned that the paper's main novelty lies more in the scalable recipe and data construction pipeline than in a fundamentally new training method. The rebuttal did a good job clarifying leakage issues, retrieval setup, and the empirical value of the recipe, and three reviewers indicated that their concerns were resolved. Still, because some novelty/positioning concerns remained and the strongest claim should be framed carefully, I view the paper as better suited for weak accept than full accept. I therefore recommend weak accept.